# Zinc-dependent regulation of zinc import and export genes by Zur

Seung-Hwan Choi[1,*], Kang-Lok Lee[1,*], Jung-Ho Shin[1], Yoo-Bok Cho[1], Sun-Shin Cha[2] & Jung-Hye Roe[1]

In most bacteria, zinc depletion is sensed by Zur, whereas the surplus is sensed by different regulators to achieve zinc homeostasis. Here we present evidence that zinc-bound Zur not only represses genes for zinc acquisition but also induces the *zitB* gene encoding a zinc exporter in *Streptomyces coelicolor*, a model actinobacteria. Zinc-dependent gene regulation by Zur occurs in two phases. At sub-femtomolar zinc concentrations (phase I), dimeric Zur binds to the Zur-box motif immediately upstream of the *zitB* promoter, resulting in low *zitB* expression. At the same time, Zur represses genes for zinc uptake. At micromolar zinc concentrations (phase II), oligomeric Zur binding with footprint expansion upward from the Zur box results in high *zitB* induction. Our findings reveal a mode of zinc-dependent gene activation that uses a single metalloregulator to control genes for both uptake and export over a wide range of zinc concentrations.

[1] Laboratory of Molecular Microbiology, School of Biological Sciences, and Institute of Microbiology, Seoul National University, Seoul 151-742, Korea. [2] Department of Chemistry and Nano Science, Ewha Womans University, Seoul 03760, Republic of Korea. * These authors contributed equally to this work. Correspondence and requests for materials should be addressed to J.-H.R. (email jhroe@snu.ac.kr).

Transition metals such as iron, zinc, copper and manganese are key constituents of the cell but are toxic when in excess. In bacteria, their intracellular 'free' levels are maintained within a narrow range[1–3]. This homeostasis is achieved primarily through regulating transcription of genes for metal acquisition, utilization, trafficking and exporting by specific metal-sensitive regulators[4,5]. Almost all metal acquisition genes are regulated by repressor-type regulators combined with cognate metals as co-repressors. Depletion of the specific co-repressor metals induces (derepresses) acquisition genes. On the other hand, metal efflux/ sequestration genes are induced by specific metals, which act as co-activators for activator proteins or as inducers for repressor proteins[6]. In most systems reported so far, the depletion or surplus of each specific metal are sensed by separate regulator proteins to achieve homeostasis for the specific metal. However, in *Xanthomonas campestris*, there is a report that Zur activates a putative zinc-export gene, while repressing putative zinc-uptake genes[7].

Zinc is an abundant transition metal that serves catalytic, structural, redox-modulatory and regulatory roles[8]. Its high binding affinity, next to copper among Irving-Williams series of metals, can cause competitive mismetallation, toxifying cells when in excess[3,9]. Bacteria encounter a wide fluctuation of zinc concentrations in the environment. Zinc chelation and sequestration by competing microbes or host cells easily create zinc deficiency[10], whereas nutrient-rich and metal-rich environments in animal guts and soils present toxic amounts of zinc. Zinc depletion is mostly sensed by Zur, a member of ferric uptake regulator (Fur) family proteins, conserved across major bacterial phyla such as proteobacteria, firmicutes, cyanobacteria and actinobacteria[11]. Zinc-bound Zur acts as a repressor for genes encoding high-affinity zinc importer (*znuABC*) and ribosomal proteins to mobilize zinc[12]. Zinc surplus is sensed by various regulators such as ZntR of the MerR family and CzrA/SmtB/AztR of the ArsR family to induce zinc-efflux genes[5]. In *Escherichia coli*, where zinc homeostasis is best studied, the uptake regulator Zur was reported to respond to low level of 'free' cytoplasmic buffered zinc, in femtomolar range[13], whereas the efflux regulator ZntR responds to free zinc of femtomolar to nanomolar range[13,14]. In *S. coelicolor* and *B. subtilis*, Zur was also reported to respond to femtomolar zinc, to control zinc uptake/ mobilization genes[15,16].

In Gram-positive antibiotic-producing *Streptomyces coelicolor*, cellular differentiation (sporulation) and antibiotic production is critically affected by zinc and perturbation of zinc homeostasis[17–19]. Zinc acquisition genes encoding high-affinity ABC transporter and its homologue (*znuABC* and *znuB2C2*), cysteine-less ribosomal protein paralogs to mobilize zinc (*rpmG/B* and *rpmF*), and a zincophore called coelibactin (SCO7676-92) are repressed by zinc-bound Zur[17,19,20]. Structural studies revealed two regulatory zinc-binding sites in addition to a structural one in Zur[15]. Existence of two regulatory sites with slightly different zinc-binding affinity was proposed to enable graded expression of Zur target genes in response to zinc availability in femtomolar concentration ranges[15]. To understand the function of Zur in its entirety, a genome-scale assessment of its direct target genes was in need.

Here, we use ChIP-chip analysis to identify a large number of Zur-binding sites in the chromosome, and find a zinc-dependent gene encoding a zinc-exporter that is regulated by Zur over a wide range of zinc concentrations. We investigate the zinc-dependent modulation of Zur activity from femtomolar to micromolar zinc, showing a dual role as a repressor and an activator. This study reveals the mode of achieving zinc homeostasis by Zur that binds to both import and export genes at the Zur-box sequences, and exerts a repressor or an activator function over a wide range of zinc concentrations.

## Results

### Zur is an abundant protein with extensive binding sites.

Before analysing Zur-regulated genes, we determined the amount of Zur protein in the cell under varying zinc concentrations. Cells were either untreated or treated with chelator TPEN (5.0–5.9 μM) or $ZnCl_2$ (100 μM) for 1 h before obtaining cell extracts. Western blot analysis revealed that the amount of intracellular Zur stayed nearly constant throughout these treatments (Fig. 1a). Compared with known amounts of purified Zur, the intracellular Zur was determined to be ∼3.5 ng in 5 μg total proteins in each cell sample. This corresponds to ∼3.7 μM Zur, based on the assumption that ∼43% of dry cell weight is from the protein, and the wet cell weight is ∼5.6-fold of the dry weight[21], and that the cell density is 1. This analysis revealed that Zur is an abundant protein, and its level does not fluctuate on changes in zinc concentration over a wide range.

The binding sites of Zur throughout the genome were determined by ChIP-chip experiments as described below. Regions of the genome significantly enriched by Zur binding were identified at 172 positions. They were ranked by relative peak intensity, the average of the $\log_2$ ratios of the 10 highest consecutive probes in each selected region (Supplementary Table 1). These sites encompassed all the previously determined Zur-regulated promoters; *znuA*, *znuB2/C2*, *rpmF*, *rpmG/B*, and promoters of SCO7676 and 7,681/7,682 in a gene cluster for synthesizing enterobactin-type zincophore[17,19,20] (Fig. 1b). MEME (multiple EM for motif elicitation; http://meme-suite.org/) analysis of these Zur-enriched regions revealed a 15-bp Zur-binding motif tGaNNatSatNNtCa, which can be viewed as a 7-1-7 palindrome (Fig. 1c). It is an improved version of the consensus Zur-box motif from previous ones determined from three to six Zur-binding sites[17,19] (Supplementary Fig. 1). This sequence shares some features within the central 15-bp of the computational Zur-box motif (21-bp palindrome) obtained from the *znuA* genes of 17 actinobacterial genomes, taaTGaNAANNNTTNtCANta (ref. 22). In 169 out of 172 sites, the Zur-box motif was located within 100 bp from the peak midpoint, indicating that the highly represented Zur-binding sites exhibit pronounced sequence-specificity. Among the 172 sites, 113 were located within 500 bp upstream of an ORF, although 72 out of which resided also within the coding region of a neighbouring ORF. Only 41 sites were genuinely located in the intergenic region.

### Zur activates *zitB* encoding a putative zinc exporter.

Among the top 1% Zur-binding sites, we identified a candidate member of Zur regulon (SCO6751), which encodes a putative metal exporter of the cation diffusion facilitator (CDF) superfamily (Fig. 1b). When compared with other known CDF-type zinc exporters, as well as another putative CDF family exporter (SCO1310) encoded in the *S. coelicolor* genome, SCO6751 was grouped closely with *E. coli* zitB and *czcD* genes from *B. subtilis*, *Xanthomonas campestris*, *Mycobacterium smegmatis* and *Corynebacterium glutamicum* (Supplementary Fig. 2). Based on sequence similarity, metal-transport function, and zinc-specific gene induction (see below), we named SCO6751 as *zitB*. The SCO1310 gene encoding another putative CDF family exporter was closely clustered with the *zitA* from *Mycobacterium tuberculosis* (Supplementary Fig. 2).

The *zitB* (SCO6751) gene is most likely transcribed as a monocistronic unit. The Zur binding was detected as a broad peak, which centred upstream of the coding region in the ChIP-chip analysis (Supplementary Fig. 3). To verify its regulation by zinc and Zur, we monitored *zitB* transcripts from the wild type and *Δzur* cells treated with various divalent metal ions or zinc-chelator TPEN for 30 min. S1 mapping analysis demonstrated that it is induced specifically by Zn(II), and the induction is dependent on Zur, which functions as an activator (Fig. 1d). Zinc

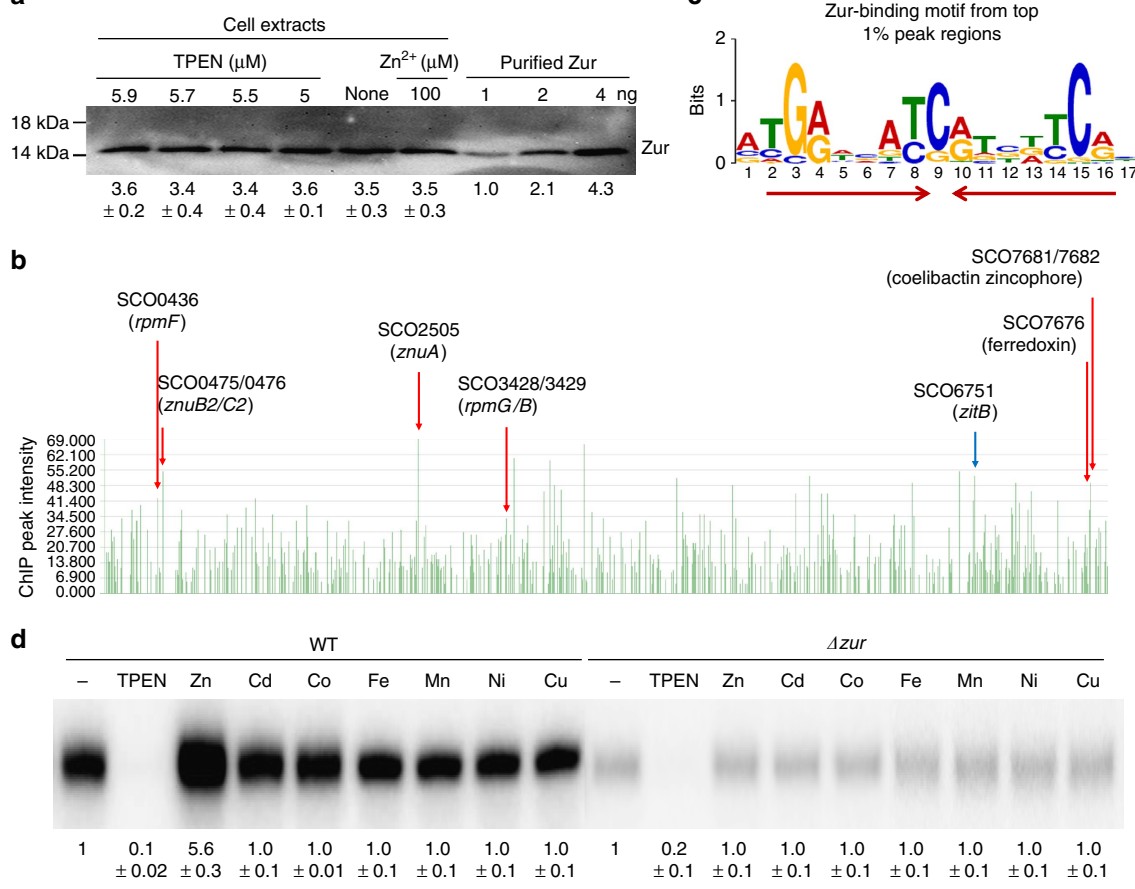

**Figure 1 | The abundance of Zur and its genome-wide binding in *S. coelicolor*.** (**a**) Analytical western blot analysis of Zur. Exponentially grown *S. coelicolor* M145 cells were either untreated or treated with varying concentrations of chelator TPEN (5.9, 5.7, 5.5 and 5.0 μM) or 100 μM ZnSO₄ for 1h before cell harvest. Crude cell extracts were analysed by western analysis, in parallel with quantified amount of purified Zur (1, 2 and 4 ng), using polyclonal antibodies against Zur. The amount of Zur in each loaded sample was estimated in ng, taking the band intensity of 1ng purified Zur as 1.0. Average values with s.d.'s from three independent experimental samples were presented. (**b**) Zur-binding peaks throughout the whole genome from ChIP-chip analysis. The peak intensity values (*y* axis) were calculated from the average of the log₂ ratios of 10 highest consecutive probe signals for each Zur-enriched site. Known promoter sites of Zur-repressed genes were indicated with red arrows. A new promoter site with Zur-binding consensus sequence was indicated with a blue arrow. (**c**) The Zur binding motif was extracted from the highly enriched 172 Zur-binding regions by multiple EM for motif elicitation (MEME), with E-value of 3.9e-233. (**d**) The zinc-specific and Zur-dependent induction of the *zitB* gene. Transcripts from SCO6751 (*zitB*) gene were analysed by S1 mapping. Exponentially grown wild type (WT) and Δ*zur* mutant cells were treated with 6 μM TPEN or various metal salts (ZnSO₄, CdCl₂, CoSO₄, FeSO₄, MnCl₂, NiSO₄ and CuSO₄) at 100 μM for 30 min before cell harvest. The amount of *zitB* transcript was quantified and presented in relative value with that in non-treated sample as 1.0. Values from three independent experiments were presented as average with s.d.'s. The *P* values for all the measurements in TPEN and zinc treatment to WT and TPEN treatment to Δ*zur* mutant were <0.001 by Student's *t*-test.

chelation by TPEN decreased its expression, and other divalent metal salts of Co(II), Cd(II), Fe(II), Mn(II), Ni(II) and Cu(II) at 0.1 mM did not induce *zitB* expression significantly (Fig. 1d). In the Δ*zur* mutant, the basal level of *zitB* expression under non-treated condition decreased to ∼20% level of the wild-type value, indicating the contribution of Zur in activating *zitB* expression. TPEN further decreased the *zitB* mRNA level, suggesting that there may be some additional regulation other than Zur that acts on the *zitB* promoter under extreme metal depleting condition.

The physiological function of ZitB was examined by overexpressing it on pSET-based integration vector. The strong *ermE* promoter-driven overexpression of *zitB* in the wild-type *S. coelicolor* caused a defect in sporulation, showing a white phenotype, and reduced antibiotic production on R2YE plates (Fig. 2a). This coincides with previous observations that a defect in zinc homeostasis inhibits differentiation and antibiotic production[17–19]. In *zitB*-overexpressing cells the level of *znuA* mRNA, which is derepressed upon zinc starvation, was elevated

by ∼5-fold in comparison with the wild type (Fig. 2b). This suggests that *zitB* overexpression caused zinc depletion in the cell. We then measured the total amounts of zinc and other divalent metals in the cell by inductively coupled plasma - mass spectroscopy (ICP-MS) analysis. Results in Fig. 2c demonstrated that the *zitB* overexpression caused a marked reduction in the amount of total zinc and iron from 124,432 to 29,712 and 114,771 to 32,391 p.p.b., respectively. It also caused reduction in the less abundant cobalt and nickel from 2,384 to 812, and from 4,204 to 1,030 p.p.b., respectively. On the other hand, the amount of copper did not change. Cobalt and manganese were not detectable. These results indicate that ZitB indeed functions as an exporter for zinc, and for nickel and cobalt as well.

**Zur activates the *zitB* expression in a biphasic manner.** As Zur controls both the uptake (*znuA*) and the export (*zitB*) genes for zinc, we compared the zinc responsiveness of Zur in regulating

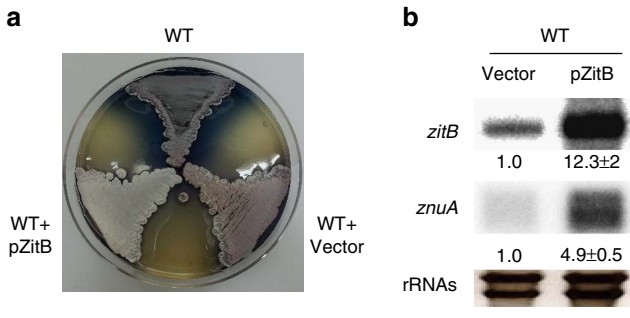

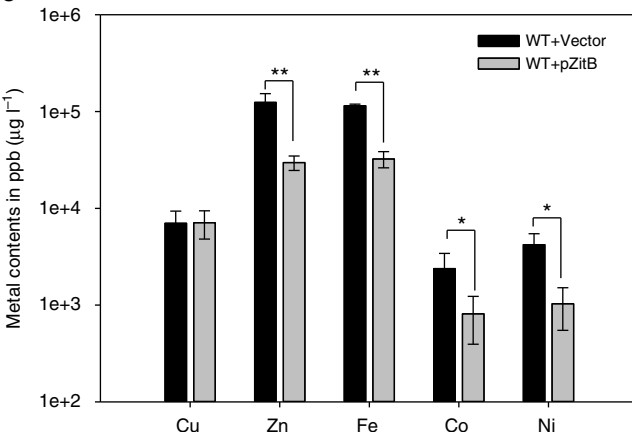

**Figure 2 | Overexpression of *zitB* hinders differentiation and causes a decrease in the content of Zn as well as Fe, Co and Ni.** (**a**) The *ermEp::zitB* construct on pSET162 plasmid (pZitB) was introduced into the chromosome of the wild-type *S. coelicolor*. The pZitB containing strain showed defect in sporulation (white phenotype) and antibiotic production, whereas the wild-type strain with or without the empty vector (pSET) demonstrated grey spore formation and blue antibiotic production. (**b**) Expression of the *zitB* and *znuA* genes in pZitB containing strain. The *zitB* and *znuA* transcripts were analysed by S1 mapping. The band intensity was quantified and presented as relative values obtained from three independent experiments and measured with *P* values of <0.001 by Student's *t*-test. (**c**) Intracellular amounts of divalent cations (Cu, Zn, Fe, Co and Ni) in the wild-type cells with empty pSET162 vector or pZitB, as assayed by ICP-MS analysis. Mn and Cd were not detected. Average values from three biologically independent samples were presented in ppb (μg per litre wet mycelium) with error bars representing s.d.'s. * and ** indicate measurements with *P* values of <0.05 and 0.001, respectively, by Student's *t*-test.

both genes. *S. coelicolor* cells exponentially grown in yeast extract-malt extract (YEME) media were treated with TPEN (5–6.5 μM) or $ZnSO_4$ (up to 150 μM) for 30 min, followed by S1 mapping of the *znuA* and *zitB* transcripts. Consistent with the previous report[19], the *znuA* transcripts increased from nearly none to full expression level, as TPEN increased from 5 to 6.5 μM (Fig. 3a). When treated with TPEN, the *zitB* transcripts decreased as TPEN increased, in an opposite direction to the *znuA* expression. The small amount of *zitB* expression in non-treated YEME further increased by adding $ZnSO_4$ up to 100 μM (Fig. 3a). Induction and repression patterns of *zitB* and *znuA* were confirmed by quantitative reverse transcription-PCR (qRT-PCR; Supplementary Fig. 4).

The relative expressions of *znuA* and *zitB* genes were plotted against the amount of zinc in the medium, by taking the maximally expressed level as 1.0 (Fig. 3b). The available zinc concentrations in the presence of TPEN were calculated by using the web-based solution tool WEBMAXC (http://web.stanford.edu/~cpatton/

webmaxc/webmaxcS.htm). YEME media, in the absence of added metal, contained 1.33 μM $Zn^{2+}$ when analysed by ICP-MS. The zinc-dependent *zitB* induction demonstrated a biphasic curve. At low zinc in the sub-femtomolar range, where the *znuA* expression changes from the full (1.0) to near zero level, the *zitB* increased from nearly zero to ~10% of the maximally induced level (phase I). At high zinc of over 10 μM, the *zitB* expression increased markedly, reaching the maximal level at ~100 μM (phase II). Between 0.6 fM and 1.3 μM zinc, which correspond to 5 μM TPEN-treated and non-treated conditions, respectively, the *zitB* level stayed nearly constant. To compare the zinc-sensitivity of Zur regulation between the *znuA* and *zitB* genes in the sub-femtomolar range, we re-plotted the relative expression of the *zitB* gene in phase I by taking the non-treated or 5 μM TPEN-treated level as 1.0. The data (Supplementary Fig. 5) demonstrated that, within this low zinc range, both the *znuA* and *zitB* expressions show nearly identical zinc responsiveness. This suggests that under low zinc condition, Zur binds to both genes with similar sensitivities to zinc, whether it acted as an activator or a repressor.

**Location of Zur binding in the *zitB* promoter.** To understand how Zur positively regulates *zitB*, the transcription start site (TSS) of the *zitB* gene was determined by high-resolution S1 mapping, following induction with 100 μM $ZnSO_4$ for 1 h. The TSS (+1) was located at the C residue 50 nt upstream from the initiation codon (Fig. 4a). The −10 (TTGACT) and −35 (TTGCCC) elements of the *zitB* promoter were predicted, and a Zur-box motif was localized 8 nt upstream from the −35 hexamer (Fig. 4b). The position of the Zur-box sequence was compatible with the role of Zur as an activator, not to inhibit RNA polymerase binding.

We then performed DNase I footprinting of Zur binding on the *zitB* promoter DNA by capillary electrophoresis. Increasing amounts of Zur were incubated with 267 bp *zitB* DNA (from −228 to +39 nt relative to TSS) in the presence of 75 μM $ZnSO_4$. When compared with the DNA-only sample, the binding of Zur at 0.45 μM was enough to protect a region between −78 and −40 nt that harbours the Zur-box motif (Fig. 4c). Zur at higher concentrations (up to 9 μM) protected further upstream region, up to −138 nt from the TSS, expanding the size of the footprint from 38 to 98 bp-long.

**Zur binds similarly to the Zur-box DNAs of *zitB* and *znuA*.** Since the zinc responsiveness of the *zitB* induction in phase I was similar to that of the *znuA* repression, we determined the binding affinity of Zur to both genes. Electrophoretic mobility shift assay (EMSA) experiments were done with increasing amounts of Zur on 25 bp of Zur-box DNA of each gene (Supplementary Fig. 6a). Since the *znuA* gene contains two Zur-box sequences, we prepared both site 1 and site 2 DNA probes of 25 bp each. The EMSA assay at 5 μM $ZnSO_4$ with Zur from 0 to 90 nM showed that Zur bound to *znuA*-1, *znuA*-2 or *zitB* Zur-box DNA probes of 25 bp with comparable affinities. The shifted position of Zur-DNA complex band corresponds to the binding of a dimeric Zur on each DNA probe. The binding curve of the EMSA data demonstrated that Zur bound to the *znuA*-1 site with the highest affinity (Kd ~15 nM), whereas the binding affinity to *znuA*-2 and *zitB* was nearly identical with Kd of ~19 nM (Supplementary Fig. 6b). The similarity in Kd values is consistent with the observation that Zur-binding was enriched with similar peak intensities at the *znuA* and *zitB* sites in ChIP-chip analysis (Supplementary Table 1).

**High Zn extends footprints by oligomeric Zur binding.** Since the amount of Zur protein stayed relatively constant in the cell, and since the *zitB* induction increased markedly at >10 μM zinc

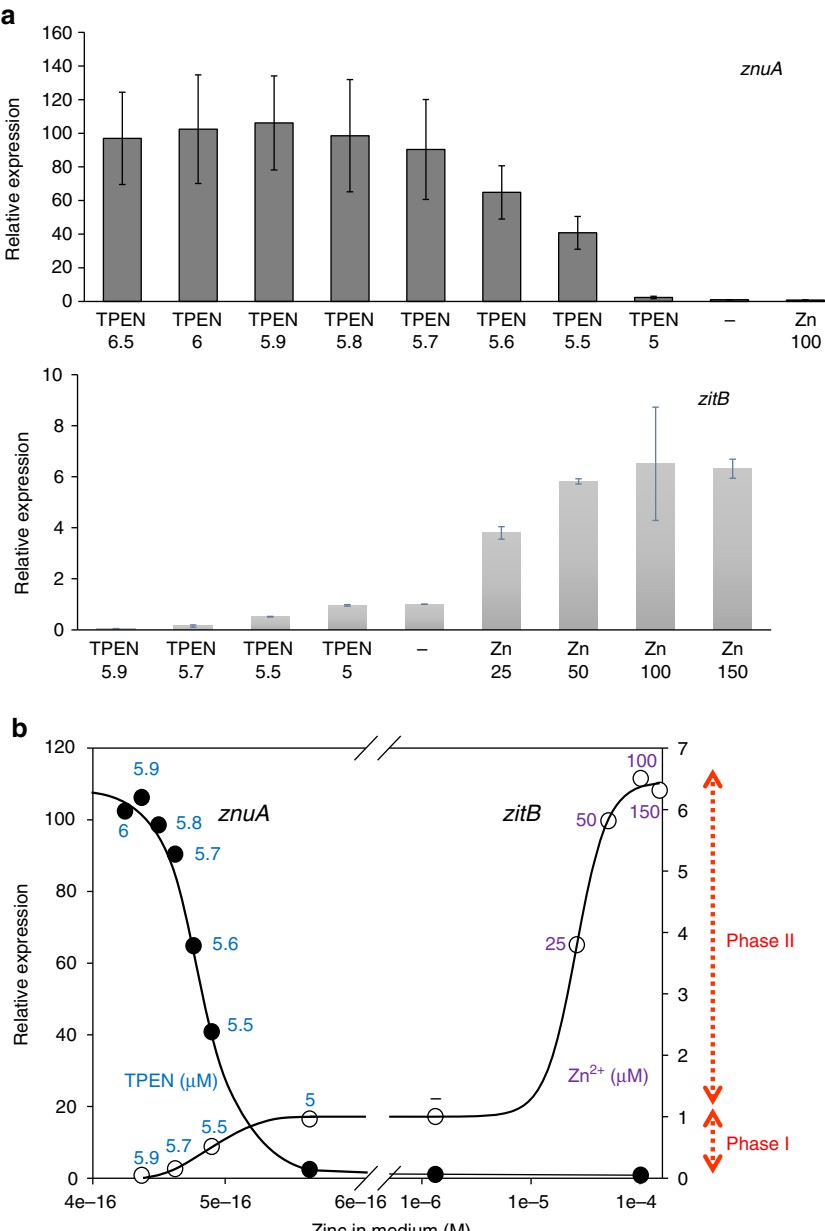

**Figure 3 | Zinc-responsive expression of the *zitB* gene in comparison with the zinc uptake (*znuA*) gene.** (**a**) S1 mapping analysis of *znuA* and *zitB* transcripts under TPEN or zinc-treated conditions. The wild-type cells grown in YEME medium to exponential growth (OD$_{600}$ of 0.4 to ~0.5) were treated for 30 min with TPEN (5.0–6.5 µM as indicated), ZnSO$_4$ (25–150 µM as indicated) or none, before cell harvest. Quantifications of S1 mapping results were done from 11 independent experiments for *znuA*, and 3–6 experiments for *zitB* transcript analysis. The relative expression values with s.d.'s were presented, taking the untreated sample values as 1.0. The *P* values for all the measurements were <0.001 by Student's *t*-test, except the *zitB* value for 5 µM TPEN treatment (*P* > 0.8). (**b**) The relative expression levels of *znuA* and *zitB* mRNAs were plotted against the concentrations of zinc in the medium. The maximally expressed levels were drawn as equal heights in the *y* axis for both *znuA* (solid circle; left axis) and *zitB* (open circle; right axis). The concentration of zinc in the TPEN-treated samples was calculated from the concentrations of treated TEPN (µM; in blue number). The added amount of Zn (µM) was indicated in purple. The amount of zinc in non-treated YEME medium ( − ) estimated by ICP-MS was 1.33 µM. The biphasic induction of the *zitB* gene was labeled as phase I (at sub-femtomolar zinc) and phase II (at >micromolar zinc).

(phase II), we examined the Zur-DNA binding under varying zinc concentrations. DNase I footprinting on *zitB* DNA was performed with fixed amount of Zur (2.7 µM) and varying ZnSO$_4$ from 2.5 to 10 µM. Results in Fig. 4d demonstrated that the Zur-footprint extended further upstream as zinc increased, up to − 138 nt from TSS. We present sequence information of the *zitB* promoter and its upstream region in Supplementary Fig. 7. This expansion in Zur-binding region is likely to lie behind the induction of *zitB* by

zinc in phase II. It can be postulated that high concentrations of zinc could have caused some changes in Zur and its DNA-binding behaviour. We examined whether high zinc induces Zur oligomerization in the absence of DNA. Addition of up to 0.5 mM ZnCl$_2$ to 10 µM Zur did not cause any change in mobility on 5% native polyacrylamide gel electrophoresis (PAGE; Supplementary Fig. 8). Therefore, Zur appears not to form oligomers beyond dimer by itself under high zinc condition.

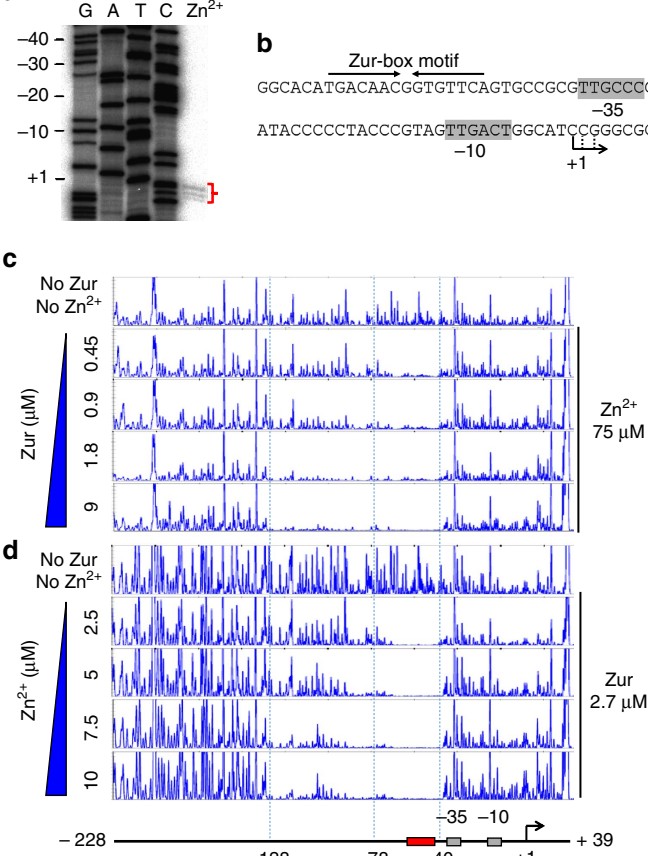

**Figure 4 | Footprinting analysis of Zur binding to the *zitB* promoter region.** (**a**) Determination of the *zitB* TSS by high-resolution S1 mapping. Exponentially grown cells were treated with $ZnSO_4$ at 100 μM for 1 h before RNA preparation. The 5′ end position of the *zitB* transcript (+1) was determined by electrophoresis of S1-protected DNAs on a 6% polyacrylamide gel containing 7 M urea, along with sequencing ladders generated with the SCC5F2A cosmid DNA and the same primer used to generate the S1 probe. The position of the longest protection was assigned as +1. (**b**) The positions of the predicted −10 and −35 elements of the *zitB* promoter and the Zur-box motif. (**c,d**) DNase I footprinting patterns of Zur-*zitB* DNA interaction as analysed by capillary electrophoresis. (**c**) Footprinting under varying Zur protein concentrations. The DNA probe containing the *zitB* gene from +39 to −228 nt relative to the TSS (+1) was incubated with increasing amounts of Zur (0.45, 0.9, 1.8 and 9.0 μM) in the presence of 75 μM $ZnSO_4$. The DNA probe only with no added Zur nor zinc was analysed in parallel. The primary protection from −40 to −78, and the extended footprint at higher Zur up to −138 were indicated with dotted lines. (**d**) Footprinting under varying zinc concentrations. DNase I footprinting analysis was done with the same DNA probe, but with fixed amount of Zur at 2.7 μM, and increasing amount of $ZnSO_4$ (2.5, 5.0, 7.5 and 10.0 μM) in each binding reaction. Zinc-dependent extension of Zur footprint on the *zitB* promoter was shown.

As an initial step to unravel any changes that zinc might have caused to Zur and its binding to *zitB* DNA, we performed EMSA analysis with 33 bp *zitB* DNA containing the Zur-box motif in the middle, in comparison with the 25 bp probe. At sufficiently high concentration of zinc (20 μM) and Zur (90 nM), only a single complex band is formed on the 25 bp DNA probe (Fig. 5a, lane 2). However, on the 33 bp DNA, longer by 4 nt at each end than the 25 bp probe, two retarded bands appeared as zinc increased (Fig. 5a, lanes 4–11). We estimated the molecular

weights of retarded bands by native PAGE with different percentages of polyacrylamide. The upper band on 33 bp probe matched the mobility of tetrameric Zur-DNA complex, most likely as a dimer of dimeric Zur, and the lower band corresponded to a dimeric Zur-DNA complex (Supplementary Fig. 9). We then examined Zur binding to 46 bp DNA (from −80 to −35 nt) by EMSA. The lower band on 46 bp probe corresponded to dimeric Zur-DNA complex, whereas the mobility of the upper band matched the molecular weights of either tetrameric or hexameric Zur-DNA complex within experimental error, as judged by native PAGE with different acrylamide percentages (Supplementary Fig. 10). These results indicate that Zur has the ability to bind to *zitB* DNA as an oligomer, which is facilitated by zinc at high concentrations.

Finally, we examined Zur binding by EMSA to a long 114 bp DNA probe (−148 to −35 nt) that encompasses the entire footprinted region (−138 to −40). As shown in Fig. 5b, at a fixed amount of Zur (90 nM) and zinc at 10 and 20 μM (lanes 6 and 7), a further retarded band, whose mobility corresponded to either hexameric or octameric Zur-DNA complex (Supplementary Fig. 9), appeared. On the same 114 bp DNA probe, when a higher amount of Zur was present at 900 nM, even at 10 μM zinc, super-shifted bands of highly retarded mobility were observed (Supplementary Fig. 11a). Estimation of the extent of oligomerization was beyond the resolution limits by gel electrophoresis.

Interestingly, the footprint by Zur extended unidirectionally towards the upstream from the Zur-box as zinc increased (Fig. 4). Considering the symmetrically dimeric structure of Zur, it can be postulated that some sequence element in the upstream region may have contributed to oligomeric (or multiple) Zur binding. We examined Zur binding to DNA sequences upstream or downstream of the Zur box, using DNA probes from −59 to −148 nt (up-probe) or from −29 to +50 nt (down-probe). The EMSA results showed that some oligomeric Zur binding was observed on the up-probe as zinc increased, whereas no binding was observed on the down-probe at all zinc concentrations (Supplementary Fig. 11b,c). This clearly indicates that there is some sequence feature in the upstream of the Zur box of *zitB* promoter that facilitates oligomeric Zur binding.

**Zinc-dependent *zitB* activation via upstream sequences.** To confirm Zn-dependent Zur binding *in vivo* to the *zitB* promoter upstream region in the chromosome, we performed chromatin immunoprecipitation with anti-Zur antibody. *S. coelicolor* cells treated with 0.1 mM $ZnSO_4$ were examined along with non-treated wild type and *Δzur* cells. About 500 bp region of the *zitB* gene from −348 to +155 nt (relative to TSS) was screened by qPCR, by using 8 sets of overlapping primers, each pair of which producing 70–90 bp PCR products that encompass the whole region. ChIP-qPCR results (Supplementary Fig. 12) indicated that Zur binding occurred preferentially to the promoter upstream region rather than to the downstream region. It also showed that 0.1 mM zinc addition substantially increased Zur binding, compared with the non-treated (∼1 μM zinc in the media) sample.

The effectiveness of the *zitB* upstream region in zinc-dependent gene activation was examined *in vivo* by using a heterologous reporter gene encoding β-glucuronides (GUS). Recombinant p*zitB*-GUS fusion plasmids that contain the *zitB* regulatory region up to −60 or −228 nt were constructed on pSET152-based integration vector (Fig. 5c). Following chromosomal integration through the *att* site, the reporter gene expression was assessed after zinc treatment. The S1 mapping of GUS transcripts demonstrated that the *zitB* regulatory region with the Zur-box motif only (up to −60) allowed only marginal

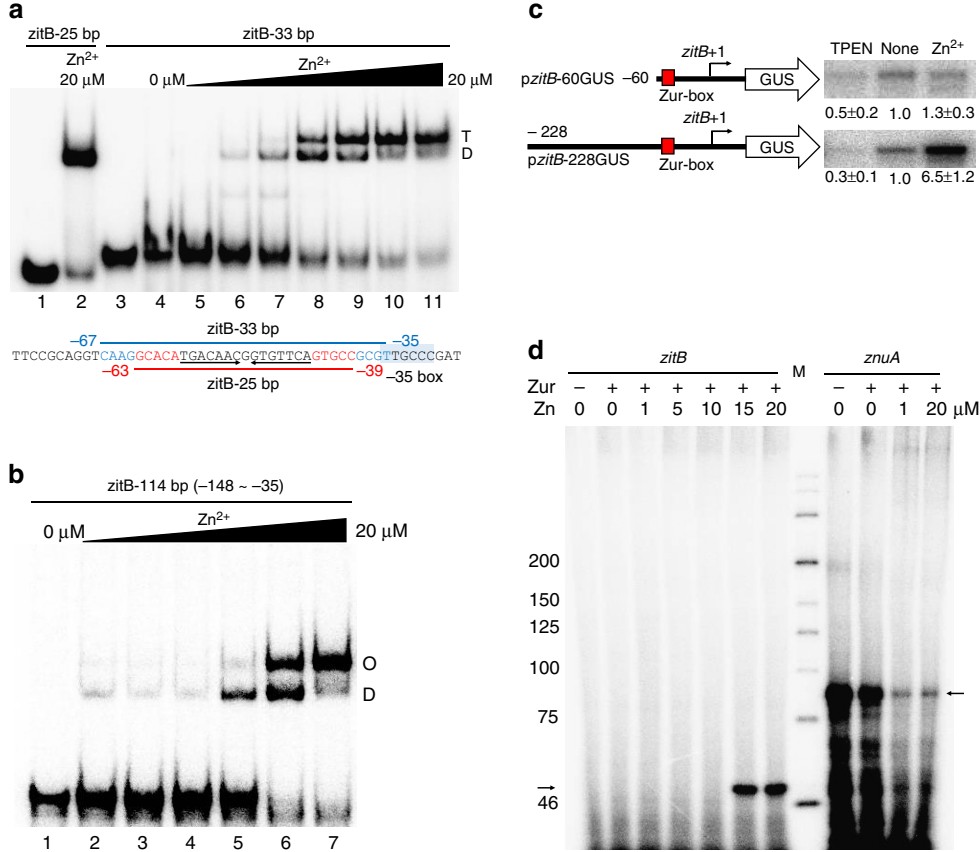

**Figure 5 | Zinc-dependent formation of multimeric Zur-*zitB* DNA complexes *in vitro* and the contribution of Zur-box upstream region on *zitB* activation *in vivo*.** (**a**) EMSA analysis of Zur binding on 33 bp *zitB* DNA probe in comparison with the complex on 25 bp *zitB* DNA. Increasing amounts of zinc (0, 0.1, 0.5, 1.0, 2.5, 5, 10 and 20 μM) were included in the binding buffer with 90 nM Zur. The molecular weights of the retarded bands were estimated from electrophoretic mobility on native PAGE with different acrylamide percentages (Supplementary Fig. 9), and were marked as T (tetramer) or D (dimer). (**b**) EMSA analysis on the 114 bp *zitB* probe (from −148 to −35 nt). Increasing amounts of zinc (0, 0.5, 1, 5, 10 and 20 μM) were included in the binding buffer with 90 nM Zur. Based on the estimated molecular weights from native PAGE mobility, the retarded bands were indicated by O for octamer, and D for dimer. (**c**) Expression of GUS reporter gene linked with the *zitB* promoter region from +50 to −60 nt (p*zitB*-60GUS) or to −228 nt (p*zitB*-228GUS). *S. coelicolor* cells containing the chromosomally integrated reporter gene were either non-treated or treated with 10 μM TPEN or 100 μM ZnSO₄ for 30 min. Quantitation of S1 mapping results were done from three independent experiments, and the relative expression values were presented by taking the non-treated level as 1.0. The *P* values of all the relative measurements except zinc-treated p*zitB*-60GUS were ≤0.001 by Student's *t*-test. (**d**) *In vitro* transcription assays of the *zitB* and *znuA* promoters in the presence of purified Zur (50 nM) and RNA polymerase core enzyme (*E. coli*) and the housekeeping sigma factor HrdB (*S. coelicolor*). Varying amounts of ZnSO₄ (0, 1, 5, 10, 15 and 20 μM) were added in the transcription buffer. Predicted lengths of the *zitB* and *znuA* transcripts are 52 nt (left arrow) and 87 nt (right arrow), respectively.

gene activation, whereas the *zitB* upstream sequence up to −228 nt enabled full activation of the reporter gene expression (Fig. 5c).

Finally, we performed run-off transcription assay with purified Zur to examine its ability to activate *zitB* transcription *in vitro*. The *zitB* promoter sequence deviates farther from the prominent promoter consensus sequence recognizable by the housekeeping sigma factor HrdB, whereas *znuA* promoter matches well with the consensus (Jeong *et al.*[23]; Supplementary Fig. 7b). The *zitB* promoter lacks the critical A residue at position −11, and is likely to be recognized very weakly by HrdB, if not recognized by another alternate sigma factor. Transcription from the 339-bp *zitB* DNA template (−287 to +52) was allowed to occur in the presence of *E. coli* RNA polymerase core enzyme and the sigma factor HrdB of *S. coelicolor*, fixed amount of Zur (50 nM) and varying amounts of ZnSO₄ from 0 to 20 μM. Transcription from the *znuA* template (−107 to +87) was examined in parallel. Results in Fig. 5d demonstrated that Zur activated *zitB* transcription directly under high zinc (15 and 20 μM) conditions,

but not under lower zinc, supporting the proposal that the phase II activation can occur by Zur binding only. In contrast, the *znuA* gene was highly transcribed under no-zinc condition, but was repressed efficiently by 1 μM zinc, consistent with *in vivo* observations (Fig. 5d). The failure to detect *zitB* transcription under lower zinc condition could be due to non-optimal composition of RNA polymerase holoenzyme for the weak *zitB* promoter. Otherwise, there still remains the possibility that some additional factor(s) may also contribute to activate *zitB* under low zinc condition (phase I).

## Discussion

CDF family transporters are distributed across three domains of life, and contribute to metal homeostasis by extruding divalent metal ions[24,25]. In this study we found that the synthesis of ZitB, which shows close sequence similarity to *E. coli* ZitB and CzcDs from various bacteria (clade IX of the Zn transporter subfamily)[24], is induced by zinc in a Zur-dependent manner in

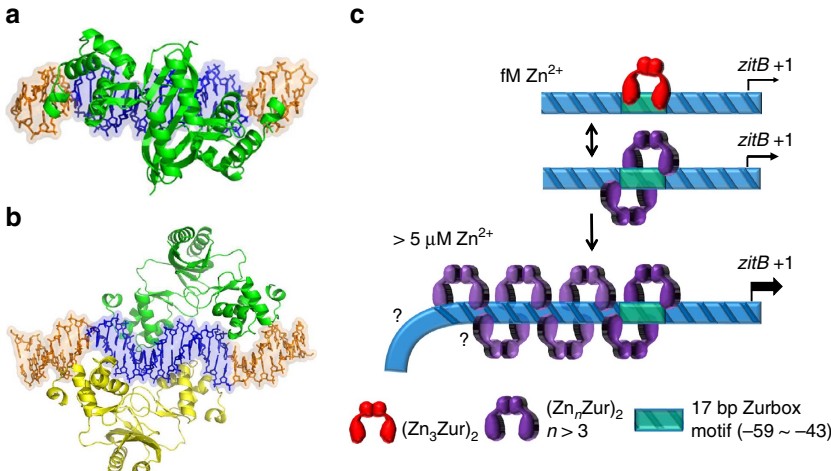

**Figure 6 | A scheme for zinc-dependent changes in the binding mode of Zur on *zitB* DNA.** A structural model for the Zur$_2$-DNA (25 bp) complex (**a**) and a model for Zur$_4$-DNA (33 bp) complex (**b**). The blue regions in DNA represent the 15 bp Zur-box motif. The two figures were prepared with the same DNA orientation, to show the distinct binding modes of dimeric versus tetrameric Zur. (**c**) A schematic model for the change in the binding mode of Zur on *zitB* promoter region as zinc level increases. The dimeric Zur with three high-affinity zinc-binding sites occupied by zinc at femtomolar range (Zn$_3$Zur)$_2$ was indicated in red, whereas the oligomers of dimeric Zur with possibly more zinc binding at low-affinity site(s) were presented in purple.

*S. coelicolor.* The cellular phenotype of ZitB overproduction indicated that ZitB extrudes zinc, as well as nickel and cobalt, but not copper (Fig. 2). The amount of total zinc in *S. coelicolor* cells grown in YEME medium was estimated to be ∼2.1 mM (Fig. 2). Considering the amount of zinc in the medium (1.33 μM), this reveals more than 1,000-fold zinc concentration in the cell. This level is higher than the 'zinc quota' of *E. coli*, yeast and some mammalian cells, estimated to be 0.1–0.5 mM (ref. 26). Therefore, *S. coelicolor* cells growing in YEME medium may already contain surplus zinc to be extruded through exporters such as ZitB.

In most bacterial systems reported so far, regulators for the uptake and export genes are different, with different metal-binding affinities[2]. An exception is *Xanthomonas campestris*, where Zur was reported to activate a putative metal-exporter gene[7]. Zur is the most prevalent regulator for zinc uptake genes in bacteria, except in *Streptococcus pneumoniae* where AdcR, a MarR family repressor, regulates zinc uptake genes[27]. In *E. coli, B. subtilis* and *S. coelicolor*, Zur responds to zinc at fM range[13,15,16]. As judged from the affinity of the specific metal that allosterically changes DNA-binding activity of the regulator (metal-responsiveness), the efflux regulators respond to zinc at femtomolar (ZntR in *E. coli*[13]), picomolar (CzrA in *S. aureus*[28]) and up to nM ranges (AztR in Anabaena PCC7120 (ref. 29)). In our study, we demonstrated that Zur responds to (binds) zinc at sub-fM range to repress zinc uptake genes fully and activate the zinc-export gene (*zitB*) partially (phase I activation). Zur further responds to μM zinc, resulting in oligomeric Zur binding to the *zitB* promoter upstream region, and activates *zitB* expression to the full level. This is a way of utilizing a single metalloregulator over a wide concentration range of the specific cognate metal. The Zur-box sequence is conserved upstream of the *zitB* homologues in other actino-mycetes. Therefore, it is likely that similar zinc homeostatic regulation may occur in other actinobacteria such as *Mycobacterium* and *Corynebacterium* spp. Sequences upstream from the Zur-box up to −138 nt did not show any pronounced conserved feature among these bacteria. Further systematic studies to identify as yet uncharacterized sequence feature within the Zur-box upstream region of *zitB* are in need.

There are numerous examples of DNA-binding proteins that can act both as a repressor and an activator, depending on its binding site relative to the promoter elements. Among metalloregulators, Fur and Zur are known to play a dual role[30–32]. Even though Fur acts mostly as a repressor of its target genes, it is reported to activate some genes by binding to the promoter upstream region in *Neisseria meningitides*[30], *Helicobacter pylori*[33] and *Salmonella enterica*[34]. In the Fur-activated genes, Fur binds to the Fur-box sequence located ∼100 to 200 bp upstream of the TSS. This far distance is not suitable to allow a direct contact between an activator and RNA polymerase on linear DNA, without a looping mechanism. Oligomeric Fur binding at high protein concentrations (in the presence of Mn$^{2+}$) *in vitro* has been proposed to repress aerobactin genes[35,36] and the *sodA* gene for MnSOD[37], and to activate the *hilD* gene encoding T3SS virulence factor by binding at −147 ∼ −219 upstream from TSS[34]. The physiological relevance of oligomeric Fur binding at high protein concentrations is not certain. Whereas the dimeric Fur can oligomerize both in the presence and absence of DNA[38,39], Streptomyces dimeric Zur in our study did not form oligomers in the absence of DNA. In the case of *X. campestris* Zur[7], the binding site upstream of the putative efflux gene showed a different sequence motif distant from the Zur-box consensus found in zinc-uptake genes. In *Neisseria meningitidis*, Zur was proposed to activate two genes encoding putative alcohol dehydrogenase and NosR-related protein, respectively, based on transcriptome analysis[31]. Meningococcal Zur was found to bind to the Zur-box sequence located at 140 nt upstream from the translational start codon of the putative alcohol dehydrogenase gene, where the binding position relative to the promoter elements is unknown[31]. The close proximity of the Zur-box in the *zitB* promoter of *S. coelicolor*, spaced 8 nt upstream from the −35 region, suggests that Zur may contact RNA polymerase to activate transcription. The possibility of the activator to recruit RNA polymerase via alpha subunits (class I activation mechanism) or to contact the domain 4 of sigma factor (class II activation), as best demonstrated for CRP/CAP in *E. coli*, can be considered[40].

What could be the mode of Zur binding on the *zitB* promoter in phase I and phase II activations? Previous studies have focused on the high-affinity zinc binding to Zur, relevant under low zinc conditions. Binding of a zinc to the 'structural' site enable the formation of dimeric Zur (Zn$_2$–Zur$_2$) that lack DNA-binding

activity. In *S. coelicolor* Zur, additional zinc ions occupy two 'regulatory' sites with sub-femtomolar sensitivities, transforming it to DNA binding-competent forms ($Zn_4$–$Zur_2$ and $Zn_6$–$Zur_2$), and enabling graded expression of its target genes in response to sub-femtomolar range of 'free' zinc[15]. In *B. subtilis,* sequential binding of zinc to one 'regulatory' site in Zur is proposed to transform it to partially ($Zn_3$–$Zur_2$) or fully active ($Zn_4$–$Zur_2$) conformation, also enabling graded gene expression of its target genes[16,41].

In phase I activation mode, where free zinc is present at femtomolar range, the three high-affinity zinc-binding sites (one structural and two regulatory) of *S. coelicolor* Zur will be fully occupied by zinc, and the functional Zur will bind to both *znuA* and *zitB* promoter sites as a dimer, fully repressing *znuA* and partially activating *zitB* expression (phase I; Fig. 3, Supplementary Fig. 5). The binding mode is likely to be the $Zur_2$-DNA complex observed on the 25 bp *zitB* DNA probe, or on longer probes under low zinc conditions (Fig. 5a,b). A molecular model of the $Zur_2$-DNA complex is presented in Fig. 6a. As the level of zinc increases to micromolar range, oligomeric Zur binding occurs. Formation of tetrameric Zur-DNA complex was captured on the 33 bp DNA probe, whose molecular structure could be modelled as in Fig. 6b. In this tetrameric Zur binding model, the specific DNA interaction appears different from the mode for dimeric Zur binding. On longer *zitB* probes at micromolar zinc, formation of hexameric or octameric Zur bindings were captured by EMSA with limited amount of Zur. From EMSA, the formation of super-retarded complex with higher concentration of Zur indicates the possibility of multimerization of Zur, with or without DNA conformational change, to underlie the activation mechanism of phase II. A schematic mode of gene activation was presented in Fig. 6c. Considering the requirement of high zinc concentrations for Zur multimerization, Zur might have additional zinc-binding sites with low zinc avidity and the occupation of those sites at high zinc concentrations might favour Zur multimerization.

In this study, we revealed a way of a specific metal to change the binding mode and activity of its cognate metalloregulator, hence regulating its homeostatic genes ranging from uptake to efflux functions. Whether and how, if any, the low-affinity zinc binding at micromolar concentrations transforms the structure of Zur to facilitate its cooperative binding to DNA as oligomers is a challenging question to solve in the future. Identification of the cryptic sequence feature that facilitates upward oligomeric binding of Zur in the *zitB* promoter is yet another interesting question to solve in the near future.

## Methods

**Bacterial strains and cultures.** *S. coelicolor* A3(2) M145 strain (John Innes Centre, UK) was used as the wild type and routinely grown in YEME liquid medium containing 10.3% sucrose. Standard culture protocols for *S. coelicolor* were followed as described[42]. To construct the ZitB-overproducing strain, SCO6751 (GI:4584498, *zitB*) was PCR-amplified and linked to the strong *ermEp* promoter[43]. The ligated 1,329-bp PCR product was cloned into pSET152-derived integration vector pSET162 (ref. 44). To construct *zitBp*::GUS reporter strain, the *zitB* promoter regions from +49 to −60 (zitB-60GUS) or to −228 (zitB-228GUS) were PCR-amplified with the zitB-60GUS forward (5′-ATATCTAGACAAA-CCGCGCCCCCAGAC-3′; the XbaI site is underlined) or zitB-228GUS forward (5′-ATATCTAGACAAACCGCGCCCCCAGAC-3′; the XbaI site is underlined) and reverse (5′-ATAGGTACCGAGAAAGCCGCCTCCTCGTG-3′; the KpnI site is underlined) primers and cloned in front of the reporter gene with SD sequence and coding region of β-glucuronides (GUS) in pSET152-derived integration vector pGUS (provided by A. Luzhetskyy)[45]. Recombinant plasmids were introduced into *E. coli* ET12567 (provided by T. MacNeil)[46] carrying pUZ8002 (provided by K. Chater)[42], followed by conjugal transfer to *S. coelicolor* M145. Apramycin-resistant exconjugants were selected, and the correct integration of *ermEp*::*zitB* or *zitBp*::GUS into the chromosome via the *att* site was confirmed by PCR and nucleotide sequencing.

**Western blot analysis.** *S. coelicolor* cells grown to $OD_{600}$ of 0.4–0.5 in YEME were sonicated in lysis buffer (20 mM Tris-HCl, pH 7.9, 10% glycerol, 5 mM EDTA, 0.1 mM DTT, 10 mM $MgCl_2$, 1 mM PMSF and 0.15 M NaCl). Protein concentration in crude cell extracts was determined by Bradford reagent solution (Bio-Rad) using BSA as a standard. Either purified Zur or cell extracts containing 5 μg proteins were electrophoresed on 13% SDS–PAGE. Polyclonal mouse antibody against Zur (prepared in our lab) and the anti-mouse IgG secondary antibody (Santa Cruz Biotechnology, SC-2005) were used at 1:5,000, 1:3,000 dilution ratio, respectively, for immunodetection (Fusion Solo; Vilber Lourmat). A representative full-sized immunoblot photo is shown in Supplementary Fig. 13. All experimental protocols that involve animals were approved by and done in accordance with the guidelines by Seoul National University Institutional Animal Care and Use Committees (SNUIACUC).

**ChIP-chip and bioinformatic analyses.** The chromatin immunoprecipitation experiment was performed for the wild type and *Δzur* mutant cells using the polyclonal antibody against Zur[47]. Exponentially grown cells were fixed with 1% formaldehyde, harvested and suspended in lysis buffer, followed by sonication. Micrococcal nuclease (100 U ml$^{-1}$) and RNase A (1 mg ml$^{-1}$) were added to further fragment the genomic DNA and degrade RNA. After removing cell debris by centrifugation, DNA fragmentation was assessed by agarose gel electrophoresis. For optimal results, samples with DNAs in the size range of ∼200–1,000 bp were immunoprecipitated with anti-Zur mouse antibody. The DNA enriched by using anti-Zur serum was labelled with Cy5 dye while DNA from the mock precipitation was labelled with Cy3. The custom-made 385 K NimbleGen microarray (Roche NimbleGen, Madison, WI) was provided by Tim Donohue (UW-Madison, WI)[48]. Sample hybridization and microarray scanning were performed by Roche NimbleGen. Regions of the genome significantly enriched for occupancy by Zur were identified using TAMALPAIS at *P* value ≤ 0.01 for a threshold set at the 99th percentile of the $log_2$ ratio of the immune-precipitated sample to the control sample for each chip[49]. Enriched regions that were significant in three biological replicates were considered. The peaks were ranked by order of intensities calculated from the average of the 10 highest consecutive probe signals for each peak. We searched for putative Zur-binding consensus sequence within the identified Zur-enriched peak regions, by using MEME (http://meme-suite.org) with palindromic ZOOPS (zero or one occurrence per sequence) constraint, and HMMER package (version 2.3.2).

**S1 nuclease mapping.** RNAs were isolated from the wild-type and *Δzur* mutant cells freshly grown to $OD_{600}$ of 0.4 to 0.5 in YEME medium, non-treated or treated with metal salts or metal chelator TPEN for 30 min. DNA probes for the *zitB, znuA* and GUS transcripts were generated by PCR from *S. coelicolor* cosmids (SCC5F2A for *zitB*; SCC121 for *znuA*; John Innes Centre) and pGUS, using pairs of 20 or 21 nt-long oligonucleotide primers whose 5′ ends correspond to −103 and +149 nt for the *zitB*, −127 and +80 for the *znuA* and −255 and +180 for the GUS gene relative to the start codon. The probe DNAs were labelled at 5′ ends with [γ-$^{32}$P] ATP and T4 polynucleotide kinase. Hybridization and S1 nuclease mapping were carried out according to standard procedures. For high-resolution mapping, the protected DNA fragments were loaded onto 6% (w/v) polyacrylamide gel containing 7 M urea, along with sequencing ladders generated from the same probe DNA. After electrophoresis, the gel was dried and exposed to a phosphor screen (BAS MP 2,040) and quantified with Fuji Phosphorimage analyzer (FLA-2,500; Fuji). Representative photos of full-sized S1 mapping autoradiographic images are shown in Supplementary Fig. 13.

**EMSA for Zur-DNA binding.** The 25 bp *zitB* probe (from −39 to −63 nt from the TSS) was generated by annealing the sense (5′-GCA CAT GAC AAC GGT GTT CAG TGC C-3′) and anti-sense (5′-GGC ACT GAA CAC CG TTG TCA TGT GC-3′) strands of synthetic oligonucleotides. The 33 bp *zitB* DNA probe (from −35 to −67) was amplified by PCR with the forward (5′-CAA GGC ACA TGA CAA CGG TGT T-3′) and reverse (5′-ACG CGG CAC TGA ACA CCG TTG T-3′) primers. The 46 bp *zitB* DNA probe (from −80 to −35) was amplified by PCR by using the forward primer (5′-CGT TTC CGC AGG TCA AGG C-3′) and the reverse primer used to generate 33 bp DNA probe. The 120 bp probe (from nt −148 to −29) was amplified by using the forward (5′-GTC GGA CCG GTC CCC CTG AC-3′) and reverse (5′-CGG GCA ACG CGG CAC TGA AC-3′) primers. The purified DNA was labelled with [γ-$^{32}$P]-ATP by using T4 polynucleotide kinase. Zur protein was purified from *E. coli* BL21 (DE3) cells (New England Biolabs) containing pET3a (Novagen)-based recombinant plasmid using Ni-NTA column[15,19]. Briefly, cells at $OD_{600}$ of 0.3 were induced with 1 mM isopropyl-β-D-thiogalactopyranoside for 6 h at 30 °C, harvested, and ruptured by Emulsiflex (Avestin) in the binding buffer (20 mM Tris-HCl, pH 7.9, 0.5 M NaCl and 5 mM imidazole). The supernatant was loaded on Ni-NTA column (Novagen). Zur was eluted with elution buffer (20 mM Tris-HCl, pH 7.9, 0.5 M NaCl and 500 mM imidazole), and dialyzed against buffer A (20 mM Tris-HCl, pH 7.8, 100 mM NaCl, 5% glycerol and 5 mM EDTA) to remove imidazole and nickel, and then buffer B (20 mM Tris-HCl, pH 7.8, 50 mM NaCl, 10% glycerol and 0.1 mM DTT) to remove EDTA. It was further dialyzed against storage buffer (20 mM Tris-HCl, pH 7.8, 50 mM NaCl, 30% glycerol and 2 mM DTT). Binding

reactions were performed with $\sim 5.5$ fmole of labelled DNA probes and 90 or 900 nM purified Zur in 20 µl of the reaction buffer (20 mM Tris-HCl, pH 7.8, 50 mM KCl, 1 mM DTT, 0.1 mg ml$^{-1}$ BSA, 5% glycerol and 0.1 µg of poly(dI-dC), with 020 µM ZnSO$_4$). Following incubation at room temperature for 20 min, the binding mixture was subjected to electrophoresis at 4 °C on a 5% polyacrylamide gel at 130 V in TB (89 mM Trizma base, 89 mM boric acid) buffer. After electrophoresis, the gel was dried and exposed to a phosphor screen (BAS MP 2,040) and quantified with a Phosphorimage analyzer (FLA-2,500; Fuji).

**In vitro transcription assay.** The housekeeping sigma factor HrdB of *S. coelicolor* was purified from *E. coli* BL21 containing the recombinant plasmid pET15b-ScHrdB that contains the entire coding region of the *hrdB* gene (1,554 bp) amplified from cosmid SC5B8 (provided by John Innes Center). Run-off transcription assay was done as described by Kang *et al.*[50], with some modifications. Template DNAs for *zitB* ( − 287 to + 52) and *znuA* ( − 107 to + 87) were prepared by PCR with the *zitB* forward (5′-GTCGACGCCACTTGCTCCCG-3′) and the *zitB* reverse (5′-CATGAGAAAGCCGCCTCCTC-3′) primers, or the *znuA* forward (5′-GCTTCAGGTTACCGGCGTGG-3′) and the *znuA* reverse (5′-CTG GAGCAGGCCGAGAGGGTG-3′) primers. Purified DNA (0.05 pmole), Zur (1 pmole), *E. coli* RNA polymerase core enzyme (1 Unit; NEB, M0550S), and sigma factor HrdB from *S. coelicolor* (1.12 µM) were incubated in 20 µl transcription buffer (20 mM Tris-HCl, pH 7.8, 50 mM KCl, 1 mM DTT, 0.1 mg ml$^{-1}$ BSA, 5% glycerol) at 30 °C for 5 min, in the absence or presence of ZnSO$_4$ (1–20 µM). Transcription reactions were allowed to occur at 37 °C for 5 min by adding unlabelled ATP, UTP and GTP to 400 µM, and CTP to 40 µM, with 5 µCi of [α-$^{32}$P] CTP (400 Ci per mmole, Amersham). Cold CTP (1.2 mM) was then added for 10 min. The reaction was terminated by adding 50 µl precooled stop solution (375 mM sodium acetate, pH 5.2, 15 mM EDTA, 0.1 mg ml$^{-1}$ calf thymus DNA), followed by ethanol precipitation. RNA samples were loaded onto 6% (w/v) polyacrylamide gel containing 7 M urea, followed by gel drying and autoradiography as done for S1 mapping analysis.

**DNase I footprinting with capillary electrophoresis.** The probe DNA was prepared by PCR using a 5′ 6-FAM labelled fluorescent forward primer. For 267 bp *zitB* probe, the forward (5′-GAC AAA CCG CGC CCC CAG AC-3′) and the reverse (5′-GAC GCC GGT ACA CAC GAG GAG-3′) primers encompassing a region from nt − 237 to + 38 (relative to TSS) were used. The PCR products were gel-purified by using the standard crush and soak method. Binding reactions were performed as in EMSA, except using 250 ng probe DNA in 40 µl reaction mixture. After 10 min incubation at room temperature, 40 µl of 5 mM CaCl$_2$ and 10 mM MgCl$_2$ was added, followed by adding 200 U of RQI RNase-free DNase I (Promega) for 1 min. The cleavage reaction was stopped by adding 90 µl stop solution (200 mM NaCl, 30 mM EDTA, 1% sodium dodecyl sulfate, 125 µg ml$^{-1}$ of glycogen), followed by DNA extraction and precipitation. Samples were analysed by ABI 3,730 DNA analyzer (Life Technologies).

**ICP-MS analysis of metals.** Wild-type cells containing parental pSET162 vector or pSET162-*ermEp::zitB* were grown to OD$_{600}$ of 0.4–0.5 in YEME medium. Harvested cells were washed three times with 10% glycerol, and subjected to metal analysis by ICP-MS (NexION 350D, Perkin-Elmer SCIEX) at The National Center for Inter-University Research Facilities (NCIRF) at SNU and Korea Basic Science Institute (KBSI).

**Modelling ScZur/DNA complexes.** There are two available structures of the Fur family proteins in complex with DNA; Fur from *Magnetospirillum gryphiswaldense* (*Mg*Fur[51]) and Zur from *E. coli* (*Ec*Zur[52]). Since the tetrameric conformation of *Mg*Fur in the complex seems to be more adequate for the formation of hexameric or octameric oligomer in the DNA-bound conditions, we used the *Mg*Fur/DNA complexes for modelling study. The structure of *S. coelicolor* Zur (*Sc*Zur) experimentally determined without DNA is nicely superposed on both *Mg*Fur and *Ec*Zur in the complexes with similar root mean square deviations for corresponding Cα atoms. The *Sc*Zur/DNA (25 bp) complex model was built up based on the crystal structure of the dimeric *Mg*Fur/DNA (25 bp) complex[51]. Since the available crystal structure of *Sc*Zur[15] does not represent the DNA-bound state, we modelled the *Sc*Zur structure with the *Mg*Fur structure in the *Mg*Fur/DNA complex as a template using SWISS-MODEL[53]. Then the modelled *Sc*Zur structure was superposed onto the *Mg*Fur structure in the *Mg*Fur/DNA complex. After changing DNA sequences matching with the *zitB* sequence, the relative positions of *Sc*Zur and DNA were refined by DISCOVERY (Molecular Simulation, Inc.). To make the *Sc*Zur/DNA (33 bp) complex model, the modelled *Sc*Zur structure was superposed onto each *Mg*Fur structure in the tetrameric *Mg*Fur (two dimers)/DNA (25 bp) complex, and the DNA of 25 base pairs was changed to the 33 base pair DNA with the *zitB* sequence. The positions of two *Sc*Zur molecules were manually modified to minimize steric clashes with DNA and subsequently the relative positions of *Sc*Zur and DNA were refined by DISCOVERY (Molecular Simulation, Inc.).

**Data availability.** The ChIP-chip data have been deposited in the NCBI Gene Expression Omnibus database with accession code GSE95760 (https://www.ncbi.nlm.nih.gov/geo/query/acc.cgi?acc=GSE95760). All other relevant data are available from the authors upon request.

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

## Acknowledgements

We are indebted to Drs Yann Dufour and Tim Donohue (UW-Madison) for providing microarrays and assistance in bioinformatics analysis. We thank E.-J. Woo for help in refining model structures. This work was supported by a grant (2014R1A2A1A01002846) from the Ministry of Science, ICT and Future Planning and a grant (2011-0031960) for Intelligent Synthetic Biology Center of Global Frontier Project to J.-H.R; S.-H.C. was supported by BK21-Plus fellowship for graduate students for Biological Sciences at SNU.

## Author contributions

J.-H.S. and J.-H.R. initiated this project. S.-H.C. and K.-L.L. devised and carried out most of the experiments. Y.-B.C. contributed some bioinformatics analyses of ChIP-chip data. S.-S.C. provided molecular modelling and insights about structural conversion. K.-L.L. and J.-H.R. wrote the paper, discussing with S.S.-C.

## Additional information

**Competing interests:** The authors declare no competing financial interests.

