## [Peer Review File · Nature Communications]

Reviewers' comments:

Reviewer #1 (Remarks to the Author):

In the manuscript "Zinc-dependent regulation of zinc import and export genes by a single sensor/regulator Zur" submitted to Nature Communications, the authors provide some of the first evidence that the Zur protein in *S. coelicolor*, which is known to mediate ligand-induced repression of gene expression, can also mediate ligand-responsive activation of transcription, and do so in a biphasic manner. This work is nicely done: the paper is well written and the results are quite important across several fields. There is one outstanding issue that requires additional data, but with that addition, this will be an outstanding paper.

The authors begin with a ChIP-chip assay to identify a novel Zur activated gene and focus exclusively on a known gene, *znuA*, which encodes a known component of zinc uptake transporters, and *zitB*, which they hypothesize is a zinc exporter based on sequence comparison and homology analysis. The authors show that zinc is able to activate *zitB* expression under the direct regulation of Zur. This is quite interesting because for the vast majority of Zur regulon genes across many organisms, Zur works as zinc-induced repressor of gene expression; there are few clear examples of genes that are directly activated by Zur.

Through an in vivo gene expression study (S1 mapping), the authors show that *zitB* induction by zinc occurs in two phases. They define phase I as the first stage of switching from zinc limiting to zinc sufficient growth conditions. In this case the change in levels of *zitB* mRNA is quite small and remains unchanged for several decades of increasing zinc concentration. As zinc concentration in the growth media is further increased from zinc sufficient to zinc excess conditions (i.e. phase II), the authors report a very large increase in *zitB* mRNA levels. At the fully induced level in phase II, the fold induction is nine times higher than the change in phase I. Using purified Zur protein in DNA-protein interaction assays (in vitro), the authors demonstrate zinc-induced binding of Zur to a Zur-box sequence upstream of the -35 element in the *zitB* promoter. At high protein and zinc concentrations they identify an extended Zur binding region that extends upstream of this -35 element. The authors conclude that Zur facilitates zinc homeostasis under zinc limiting conditions by directly activating expression of the zinc transporter encoded by *zitB*, but that Zur also facilitates zinc export by directly activating *zitB* expression when high zinc levels begin to stress cellular physiology.

I am very reluctant to ask authors for new experiments, however the surprising conclusion that Zur alone directly activates transcription in two surprisingly different decades of zinc concentration may have other explanations: several other factors including regulatory RNAs and other trans-acting factors that are indirectly controlled by Zur may be at play in phase I, or even phase II. If the author's models and conclusions are correct, they can readily rule out a host of alternative mechanisms by reconstituting the bi-phasic response in an in vitro transcription assay using purified Zur. With that addition, I think this manuscript would unequivocally report an important discovery that will be of great interest to the readers of Nature Communications.

Major points.

1. The authors use S1 mapping to measure the in vivo expression of *znuA* and *zitB*, and identified two phases zinc responsive activation for *zitB* transcription. Unfortunately this method does not show that Zur directly activates transcription of the *zitB* gene in both phases. The authors use inductive reasoning based on their in vitro DNA binding data showing that increasing Zur concentrations increase the number of Zur dimers bound to the upstream elements. In fact there is no data that shows Zur directly activates transcription in the absence of other regulators, small RNAs etc. This is not a trivial issue as the Fur operons in most organisms also control expression of non-coding regulatory RNA molecules (RhyB etc) that mediate Fur-responsive iron regulation in an indirect manner. My concern that indirect mechanisms may be at play comes from the pattern of zinc responsive DNA binding activity: it does not directly match the pattern of zinc induced

transcription in vivo, particularly the, the long gap in zinc response between phase I and phase II. The only way to demonstrate that Zur alone directly activates transcription in both phase I and II is to conduct zinc dependent in vitro transcription run off assays with purified Zur and RNAP. If the transcript of *zitB* is indeed increased in two phases (with a wide gap between them) under increasing zinc concentration, it is safe to draw the conclusion that *zitB* is directly activated by Zur alone in two distinct phases: i.e. a small degree of activation at very small increases in zinc, and major jump in stimulating transcription.

2. The authors should provide some argument about the transcriptional activation mechanism employed by *S. coelicolor*. There is a good case to be made given the location of the Zur box in the *zitB* promoter: many bacterial transcriptional activators (CAP, AraC etc) work by binding at this site of the promoter and extensive structural and mechanistic data provide an intimate mechanism of action. The authors need only a few sentences possible activation mechanisms of activators that work RNA polymerase to activate transcription.

3. There may be a problem with labeling in Figure 1 (d): the darkness of bands does not correlate well with the quantification results. For instance, the WT Cd²⁺ band is much lighter than the WT None band, although they were quantified as 1 and 1.0 ± 0.4 respectively. The WT Co²⁺ band is as bright as the WT Cd²⁺ band, but the value of WT Co²⁺ band is only 70% of the WT Cd²⁺ band. Besides these bands, the bands from Δzur are also not proportional to each other too. For instance, Δzur Cd²⁺ band is much darker than Δzur Co²⁺ band, but the value of the quantified Δzur Cd²⁺ band is only 10% higher than the quantified Δzur Co²⁺ band. Please clarify.

Minor points:

1. In the top 1% gene list obtained from the ChIP-chip assay, besides the eight genes already known to be regulated by Zur, the authors identified *zitB* and further characterized Zur activation at this promoter. It would be helpful if the authors listed other genes among on this top 1% list, as well as the in-depth analysis for their ChIP-chip results, in the supplemental data. If this is going to be published elsewhere, the authors should indicate this.

2. Since the Zur binding sequences for the known Zur regulated genes have been well-studied, the consensus sequence (old motif, motif #1) evolved from those sites could be used to compare with the new binding motif abstracted from the top 1% peak regions (motif #2), and also compared with the Zur binding site on the *zitB* promoter (motif #3). It would be helpful if the authors would show these three motifs/sequences together in the supplemental material: this allows for a better understanding for (1) the similarity of the Zur-box on the *zitB* promoter (motif #3) compared with the two motifs (motif #1 and motif #2); (2) the similarity of old motif (motif #1) and the new motif (motif #2). Doing it will help readers interpret their results better. The authors should consider showing the sequence for extended Zur binding region on the *zitB* promoter below their DNase I footprinting results and see how it compares with the consensus Zur-box.

3. The authors concluded that addition of Cd²⁺ or Co²⁺ cannot further induce *zitB*. Judging from Figure 1(d), it seems to be true for WT, but not true for Δzur - addition of Cd²⁺ or Co²⁺ is able to induce *zitB*.

4. Page 5, line 15 "top 1% of highly represented sites", please specify how those sites were ranked. How did they score the ChIP-chip data?

5. Page 7, line 1-2 please specify what other divalent metals that have been used to test if they are able to change the expression of *zitB*.

6. Page 7: the authors tested the phenotype of over expressing *ZitB*. To make their argument completed, it would be helpful to see the phenotype for Δ*zitB* and Δzur respectively under the high zinc conditions.

7. Figure 2(c), I wonder if the authors also have measured other metal contents, such as Fe, Mn and Cd? Did they show any changes when comparing WT+vector vs. WT+pZitB?

8. I recommend moving Figure S2 to Figure 4 and form another panel there, below the footprinting data. Figure S2 provides very important types of evidence: the peak on the promoter of zitB from the ChIP-chip assay, the mapping of the TSS on the zitB promoter, and the relative location of the Zur-box motif to -35, -10 region on the zitB promoter. These are important features of the major conclusions in this study.

Reviewer #2 (Remarks to the Author):

The authors report that expression of the zinc exporter, zitB, in *Streptomyces coelicolor* is induced by zinc. The authors report that Zur, a transcriptional regulatory protein can repress as well as activate genes involved in zinc-uptake and zinc-export of genes. Activation of genes occurs in two phases with phase one activation occurring at the location of the Zur- box motif, and Phase two involving Zur binding in oligomers upstream of the Zur-box motif.

The manuscript is relatively easy to follow and most experiments are described in detail. Although this paper describes a novel mode of action for Zur, I have some major concerns.

Major comments:

1. Using ChIP-Chip, the authors find ~17,000 potential binding sites for Zur. The authors choose to focus on only the top1% or 172 loci. 17,000 loci is a rather large number. Because this number seems unreasonable, I think it is necessary to include more details on how the analysis was performed to achieve a final list of 17,000 loci and subsequently, what represents the top 1% (i.e. strongest signal?). For example, the authors label the DNA from mock precipitations with Cy3. How were the mock precipitation samples used in the analysis? Were precipitations from WT cells labeled with Cy5 and compared against mock (Cy3) and zur deletion samples labeled with Cy5 and compared with mock also? How many individual precipitations were performed (any biological replicates?). Since a large number of putative loci were detected, it appears that maybe the conditions were not optimized. It is also worth mentioning what the breakdown for the locations in the genome that were found putatively bind Zur (how many of the 17,000 sites are intergenic, etc)?
2. To identify a motif for Zur, MEME is used. Parameters used for the MEME analysis should be included in the methods. The authors state that the motif that they discover is an improved one. How is it improved? Why do the authors feel it is improved? It seems odd that only 41 out of the 172 loci reside in intergenic regions. Can the authors explain?
3. How was Zur purified for EMSA?
4. It is critical that the authors include a non-specific DNA probe for the EMSA, especially since the ChIP-Chip experiments yielded ~17,000 loci that Zur could potentially bind.
5. The authors pick one gene, zitB, to investigate in more detail. This gene is intriguing because it is an exported and homologs have not been previously reported to be activated by Zur. Since the authors are making the claim that Zur can act as a repressor of genes involved in uptake as well as an activator of genes involved in export, then I think the authors should investigate other potential binding sites where Zur could be acting as an activator. It would be helpful to have global transcript data to compare with the ChIP data to identify genes that are activated vs repressed.

Minor comments:

1. The format of the references is odd. The text wraps from line to line. This needs to be fixed.
2. Figure 2, panel B. What do the pictures showing 2 bands represent? This panel or section should be labeled or explained in Figure legend.
3. I am not sure that the structural model adds anything. I would remove, especially since in order to generate the model, several parameters and modifications were required.

Reviewer #3 (Remarks to the Author):

Summary:

The manuscript by Choi et al. describes the role of the *Streptomyces coelicolor* zinc uptake regulator (ZUR) in the transcriptional regulation of *znuA*, a zinc import system, and *zitB*, a zinc export system. The authors discern two phases of ZUR-dependent transcriptional regulation. At sub-femtomolar zinc concentrations (phase I), *znuA* is up regulated and *zitB* is repressed. At over micromolar zinc concentrations (phase II), *znuA* remains (fully) repressed and *zitB* is transcriptionally activated. Subsequent analyses focus on the oligomeric binding of ZUR dimers to the upstream region of *zitB* during phase II, which leads to its transcriptional activation. The key findings in this manuscript as described by the authors are 1) the transcriptional activation of a zinc efflux system by ZUR, 2) the broad concentration range at which ZUR modulates expression of its targets and 3) the oligomerisation of ZUR dimers upwards from the *zitB* ZUR box.

Major Comments:

- The abstract, introduction and results of this manuscript are written such that the reader is under the impression that transcriptional activation by Zur is novel, particularly its role as an activator of metal efflux systems. However, the authors fail to appropriately introduce ZUR and in general the FUR family of regulators, thereby, in part, misleading the reader about the novelty of their work. There are many studies that describe activation by these types of regulators (e.g. doi: 10.3389/fcimb.2013.00059, doi: 10.1128/JB.00166-16, doi: 10.1093/nar/gkn328, doi: 10.1128/JB.01091-12). Their statement in the introduction "In all systems reported so far, the depletion.....for the specific metal." is incorrect considering findings from Zur in *Xanthomonas campestris*. In the discussion the authors briefly discuss Zur-dependent activation of a putative metal-efflux system in *X. campestris*, but then report that this form of regulation is distinct to that seen in *S. coelicolor*, as the relevant Zur binding site was distinct from those in the *X. campestris* Zur box regulating the zinc uptake systems. However, Zur-dependent activation was also observed in the meningococcus and in this study (Pawlik et al. 2012 J Bact.) the Zur binding site was homologous to that seen for the Zur binding sites identified upstream zinc uptake systems. Overall, it is not unexpected that Zur-dependent activation of a zinc efflux system has been identified and the lack of appropriate introduction of available literature exaggerates the claim of novelty in regards to this observation in the manuscript by Choi et al..
- Although the oligomerization of Zur upstream *zitB* is an interesting finding, the oligomerization of Fur along its target DNA has previously been shown (e.g. Escolar et al. J Biol Chem. 2000; Fréchon and Le Cam Biochem Biophys Res Commun. 1994; Le Cam et al. Proc Natl Acad Sci U S A. 1994). Even studies presented by the authors themselves (Shin et al. 2007) have indicated oligomerisation of *S. coelicolor* Zur on sites of transcriptional repression (EMSA and DNaseI footprint). Although analytical ultracentrifugation did not reveal in-solution, DNA-target free, oligomerisation of Zur, this was not performed in the presence of high zinc concentrations, which should be considered. Overall, the finding described by Choi et al. are not entirely novel for this family of metalloregulators.
- Greater emphasis on the mechanisms by which *zitB* is upregulated would significantly enhance this manuscript. The following questions follow up from previous findings on oligomerisation by Fur-like proteins and addressing these should be considered by the authors.

- 1) Does high zinc affect ionic strength and subsequently binding of Zur to its targets?
- 2) Does Zur undergo structural changes in a high zinc environment?
- 3) What are the characteristics of the DNA sequence upwards from the *zitB* zur box? (Are there repeat sequences such as described in Escolar et al. 2000 jbc? Are there any significant secondary structures?)

- The data in Fig. 3 should be backup by an alternative method such as qRT-PCR analysis, whereby the same RNA samples are used to analyse *znuA* and *zitB* expression. The data presented in fig 3 is from different S1 mapping analyses and the number of replicates vary greatly (n=3, 6 or 11). Data for *znuA* and *zitB* expression in Fig 3B should be corrected to untreated media (without TPEN or Zn supplementation). Standard deviations should be included in Fig 3B.
- How is it possible that *zitB* is nearly 2-fold down-regulated (0.6-fold) under TPEN stress in the delta-*zur* strain? This level of down-regulation is similar to that seen in the wt strain (0.5-fold). The claim that down-regulation of *zitB* under TPEN stress is Zur-dependent does not seem valid based on these results.
- Why were nickel and copper not tested in the Zur-dependent activation of *zitB* (Fig. 1D)?
- The *zitB* band in Cd stressed delta-*zur* cells appears a lot more intense than that seen for untreated delta-*zur*, but is not supported by the average? Furthermore, Cd appears to cause significantly down-regulation of *zitB* in the wt, but again this is not supported by the averaged numbers. The metal content of Cd, Co and Ni stressed cells should be determined by ICPMS to gain insight into their potential effect on Zn accumulation and subsequently zur-mediated activation of *zitB*, in isolation these stressed conditions are ineffectual.
- Analysis of the physiological function of *ZitB*, should also be confirmed by gene deletion. The role of *ZitB* in Co, Ni and Cd export under the relevant metal stress should be examined by ICP-MS using this mutant and compared to the wt and overexpressing strain. Only testing metal accumulation under low metal abundance may not reveal its true role in metal efflux.
- This manuscript would greatly benefit from examinations into the broader application of their findings. *S. coelicolor* is a model organism, its findings have to be discussed in broader context.
 - 1) Were other candidates of Zur-dependent transcriptional activation identified within the genome?
 - 2) Are regions with similar GC% or sequence identity found upwards from other transcriptionally activating Zur-boxes?
 - 3) Can similar targets be found in other bacterial genera/species?
- Although Supplementary figure 8 shows that the addition of zinc increases Zur binding at the Zur box, it does not show binding further upwards from the Zur box under high zinc conditions as compared to untreated conditions. Please discuss this.
- Statistical analyses are missing from most presented data. Often already in the form of standard deviation or standard error, but also when comparing different samples, e.g. t-test, ANOVA, etc.. This should be addressed for Fig. 1A, 1D, 2B, 2C, 3A, 3B, 5C, and Supplementary Fig. 3, 4B, 8.
- I recommend presenting the data as bar graphs instead of gels/blots with numbers underneath. This allows for easier interpretation of the data by the reader and statistical comparisons between samples to support some of the claims made. This concerns Fig. 1A, 1D, 2B, 3A, 5C.
- Please deposit the ChIP-chip data in GEO or other suitable databases.
- The experiments have been described in sufficient detail in the methods section.
- The manuscript contains various grammatical errors.

Specific comments:

Title: Consider rephrasing the title, e.g. "Zinc-dependent regulation of zinc import and export genes by Zur."

Abstract Page 2 Line 6: Insert "by Zur" after "activation"

Introduction Page 2 first paragraph line 20: "very tightly" and "within a narrow range" means the same.

Page 3 Line 4-6: As mentioned above, the statement "In all systems reported.....specific metal." is an overstatement.

Page 3: Include more detail about the role of fur and zur as transcriptional activators and describe oligomerisation of Fur.

Page 3 line 18: Include "the" after "ZntR of"

Page 3 line 19: Include "the" before "ArsR"

Page 4 line 5-6: This sentence "zinc-binding antibiotic.....zinc starvation." seems very random, consider removing.

Page 5 line 10: Include "the" before "cell".

Page 5 line 14-15: Define Y-axis in Fig 1B.

Page 5 line 14-15: Instead of only stating 1%, is there a possibility of grading the spread of high represented sites? Is there a p-value cut-off that can be used instead?

Page 5 line 21: This sentence requires rewriting "improved version" and "previous one" is too vague.

Page 5 line 21: Correct "share" to "shares"

Page 5 line 21-22: define "some features"

Page 6 line 3: Do these 41 loci also represent the best hits? Present p-values in a table.

Page 6 line 7: See comment "Page 5 line 14-15" regarding 1% cut-off.

Page 6 line 16: Naming a gene (zitA) of which the function has not been determined in this manuscript does not appear appropriate to me.

Page 6 line 23: See major comments regarding near 2-fold down regulation of zitB under TPEN stress in the delta-zur strain.

Page 7 line 3-17: See major comments regarding zitB gene deletion and testing of metal stress conditions.

Page 7 line 10: delete "the" before zitB-expression.

Page 8 line 2-3: "When treated with TPEN, the.....znuA expression." Considering the down-regulation of zitB in the delta-zur strain under TPEN stress, the data does not convincingly support the role of zur in down-regulation of zitB under severe zinc starvation. This has already been pointed out in the major comments.

Page 8 line 1-5: Fig. 3a is too messy, as mentioned in the major comments, these findings have to be backed up by an alternative method in which zitB and znuA transcription can be determined

from the same samples. All samples should be corrected against untreated cells.

Page 8 line 20: Please generate a single figure in which the data of Supp Fig 3 and Fig 3a can all be displayed.

Page 9 lines 5-7: Please put Kd values into context in terms of bioinformatics of the Zur box (p-values) and ChIP-chip data.

Page 11 line 10: Supplementary Fig. 5, this should refer to supp. Fig 6.

Page 11 line 13: Include "a" after "at"

Page 11 line 16: Include "a" after "when"

Page 11 line 19-20: Please expand on the multi-angle light scattering, I have not seen this described anywhere in the manuscript.

Page 12 line 18: Include "the" before "zitB"

Page 14 line 13-15: Please discuss the level of conservation to the -138 region in other actinobacteria.

Page 14 line 20: Please correct to: "..., Fur and Zur are known to play a dual role."

Page 15 line 7-10: Please include report by Pawlik et al. 2012 J Bact.

Page 17 line 6: Please define "Standard protocols"

Page 17 line 8: rephrase "strong promoter ermEp fragment".

Page 18 line 9: Include "The" before "Chromatin"

Page 18 line 15: Include "the" before "top 1%"

Page 18 line 16: Remove "by" after "0.01)"

Reviewers' comments:

Reviewer #1 (Remarks to the Author):

In the manuscript “Zinc-dependent regulation of zinc import and export genes by a single sensor/regulator Zur” submitted to Nature Communications, the authors provide some of the first evidence that the Zur protein in *S. coelicolor*, which is known to mediate ligand-induced repression of gene expression, can also mediate ligand-responsive activation of transcription, and do so in a biphasic manner. This work is nicely done: the paper is well written and the results are quite important across several fields. There is one outstanding issue that requires additional data, but with that addition, this will be an outstanding paper.

The authors begin with a ChIP-chip assay to identify a novel Zur activated gene and focus exclusively on a known gene, *znuA*, which encodes a known component of zinc uptake transporters, and *zitB*, which they hypothesize is a zinc exporter based on sequence comparison and homology analysis. The authors show that zinc is able to activate *zitB* expression under the direct regulation of Zur. This is quite interesting because for the vast majority of Zur regulon genes across many organisms, Zur works as zinc-induced repressor of gene expression; there are few clear examples of genes that are directly activated by Zur.

Through in vivo gene expression study (S1 mapping), the authors show that *zitB* induction by zinc occurs in two phases. They define phase I as the first stage of switching from zinc limiting to zinc sufficient growth conditions. In this case the change in levels of *zitB* mRNA is quite small and remains unchanged for several decades of increasing zinc concentration. As zinc concentration in the growth media is further increased from zinc sufficient to zinc excess conditions (i.e. phase II), the authors report a very large increase in *zitB* mRNA levels. At the fully induced level in phase II, the fold induction is nine times higher than the change in phase I. Using purified Zur protein in DNA-protein interaction assays (in vitro), the authors demonstrate zinc-induced binding of Zur to a Zur-box sequence upstream of the -35 element in the *zitB* promoter. At high protein and zinc concentrations they identify an extended Zur binding region that extends upstream of this -35 element. The authors conclude that Zur facilitates zinc homeostasis under zinc limiting conditions by directly activating expression of the zinc transporter encoded by *zitB*, but that Zur also facilitates zinc export by directly activating *zitB* expression when high zinc levels begin to stress cellular physiology.

I am very reluctant to ask authors for new experiments, however the surprising conclusion that Zur alone directly activates transcription in two surprisingly different decades of zinc concentration may have other explanations: several other factors including regulatory RNAs and other trans-acting factors that are indirectly controlled by Zur may be at play in phase I, or even phase II. If the author's models and conclusions are correct, they can readily rule out a host of alternative mechanisms by reconstituting the bi-phasic response in an in vitro transcription assay using purified Zur. With that addition, I think this manuscript would unequivocally report an important discovery that will be of great interest to the readers of Nature Communications.

Major points.

1. The authors use S1 mapping to measure the in vivo expression of *znuA* and *zitB*, and identified two phases zinc responsive activation for *zitB* transcription.

Unfortunately this method does not show that Zur directly activates transcription of the *zitB* gene in both phases. The authors use inductive reasoning based on their in vitro DNA binding data showing that increasing Zur concentrations increase the number of Zur dimers bound to the upstream elements. In fact there is no data that shows Zur directly activates transcription in the absence of other regulators, small RNAs etc. This is not a trivial issue as the Fur operons in most organisms also control expression of non-coding regulatory RNA molecules (RhyB etc) that mediate Fur-responsive iron regulation in an indirect manner. My concerns that indirect mechanisms may be at play comes from the pattern of zinc responsive DNA binding activity: it does not directly match the pattern of zinc induced transcription in vivo, particularly the, the long gap in zinc response between phase I and phase II. The only way to demonstrate that Zur alone directly activates transcription in both phase I and II is to conduct zinc dependent in vitro transcription run off assays with purified Zur and RNAP. If the transcript of *zitB* is indeed increased in two phases (with a wide gap between them) under increasing zinc concentration, it is safe to draw the conclusion that *zitB* is directly activated by Zur alone in two distinct phases: i.e. a small degree of activation at very small increases in zinc, and major jump in stimulating transcription.

>> We understand the concern that the reviewer raises. We performed in vitro transcription assays with *zitB* and *znuA* DNA templates, purified Zur at fixed concentrations, and zinc from 0 to 20 μM , using *E. coli* RNA polymerase core enzyme and the house-keeping sigma factor (HrdB) of *S. coelicolor*. Whereas the *znuA* promoter sequence matches well with the prominent promoter consensus recognizable by HrdB, the *zitB* promoter sequence deviates farther from the consensus, (Supplementary Fig. 7), suggesting that it may be recognized by HrdB quite weakly, or may be recognized by other alternative sigma factors. The in vitro transcription results demonstrated clearly that Zur activated *zitB* transcription by HrdB-bound RNA polymerase at high zinc (15 and 20 μM), but not at lower zinc conditions. This indicates that the phase II activation can occur by Zur binding only. In comparison, the *znuA* gene was highly transcribed under no-zinc condition, but was repressed efficiently by 1 μM zinc. We presented this result in Fig. 5d. The reason(s) for not detecting *zitB* transcript under lower zinc (phase I) conditions could be; 1) the in vitro transcription system was not optimally composed for the weak *zitB* promoter, resulting in very low level of expression, below the detection limit in our hands, and/or 2) some additional factor(s) may also contribute to phase I activation. Therefore, even though we showed direct binding of purified Zur in the *zur*-box sequence in the *zitB* promoter region under low zinc condition, we may still need to leave open the possibility of other regulators working in concert with Zur for phase I activation in phase I condition. We revised our text to describe these additional observations (lines 276-294).

In the Methods section, we added in vitro transcription assay (lines 484-500).

2. The authors should provide some argument about the transcriptional activation

mechanism employed by *S. coelicolor*. There is a good case to be made given the location of the Zur box in the *zitB* promoter: many bacterial transcriptional activators (CAP, AraC etc) work by binding at this site of the promoter and extensive structural and mechanistic data provide an intimate mechanism of action. The authors need only a few sentences possible activation mechanisms of activators that work RNA polymerase to activate transcription.

>> In the discussion section, we added sentences to describe possible activation mechanisms by binding upstream of the -35 position. Since the distance between the Zurbox motif and the -35 element of the *zitB* promoter is 8 nt, the possibility of both class I and class II activation mechanisms, where the activator contacts alpha and sigma subunits, respectively, can be considered, as best demonstrated for *E. coli* CAP/CRP protein. A recent review by Browning and Busby(2016) was provided. Lines 355-360.

3. There may be a problem with labeling in Figure 1 (d): the darkness of bands does not correlate well with the quantification results. For instance, the WT Cd²⁺ band is much lighter than the WT None band, although they were quantified as 1 and 1.0 ± 0.4 respectively. The WT Co²⁺ band is as bright as the WT Cd²⁺ band, but the value of WT Co²⁺ band is only 70% of the WT Cd²⁺ band. Besides these bands, the bands from Δ*zur* are also not proportional to each other too. For instance, Δ*zur* Cd²⁺ band is much darker than Δ*zur* Co²⁺ band, but the value of the quantified Δ*zur* Cd²⁺ band is only 10% higher than the quantified Δ*zur* Co²⁺ band. Please clarify.

>> The original gel photo we presented was not the representative photo that matches with the quantified values. Since the old data was obtained in the initial period of our research, we repeated the experiment, examining wider repertoire of divalent metals (Fe, Mn, Ni, Cu, in addition to Zn, Cd, Co). In the revision, we presented a representative photo from three independent experiments. The new Fig. 1d clearly shows that zinc is the only inducing metal, and that in delta-zur mutant the *zitB* gene is not induced by any metal.

Minor points:

1. In the top 1% gene list obtained from the ChIP-chip assay, besides the eight genes already known to be regulated by Zur, the authors identified *zitB* and further characterized Zur activation at this promoter. It would be helpful if the authors listed other genes among on this top 1% list, as well as the in-depth analysis for their ChIP-chip results, in the supplemental data. If this is going to be published elsewhere, the authors should indicate this.

>> We presented the list of top 1% peak locations (172 sites) as Supplementary Table 1.

2. Since the Zur binding sequences for the known Zur regulated genes have been well-studied, the consensus sequence (old motif, motif #1) evolved from those sites could be used to compare with the new binding motif abstracted from the top 1% peak regions (motif #2), and also compared with the Zur binding site on the *zitB* promoter (motif #3). It would be helpful if the authors would show these three motifs/sequences together in the supplemental material: this allows for a better understanding for (1) the similarity of the Zur-box on the *zitB* promoter (motif #3)

compared with the two motifs (motif #1 and motif #2); (2) the similarity of old motif (motif #1) and the new motif (motif #2). Doing it will help readers interpret their results better. The authors should consider showing the sequence for extended Zur binding region on the zitB promoter below their DNase I footprinting results and see how it compares with the consensus Zur-box.

>> We provided sequence comparison of Zur-box consensus sequences, determined from previous studies, along with the zitB Zurbox (Supplementary Figure 1). Accordingly, we elaborated description about consensus sequence (lines 90-96).

Promoter upstream sequences of the zitB gene were provided (Supplementary Fig. 7a). In addition, comparison of zitB and znuA promoters with the prominent promoter consensus determined from the TSS positions in the genome of *S. coelicolor* was provided (Supplementary Fig. 7b)

3. The authors concluded that addition of Cd²⁺ or Co²⁺ cannot further induce zitB. Judging from Figure 1(d), it seems to be true for WT, but not true for Δ zur - addition of Cd²⁺ or Co²⁺ is able to induce zitB.

The new gel photo represents the quantified values, and clearly shows that those metals do not induce zitB, as stated above.

4. Page 5, line 15 “top 1% of highly represented sites”, please specify how those sites were ranked. How did they score the ChIP-chip data?

>> We added this information in the Results (lines 85-87) as well as in the Methods section for ChIP-chip data analysis (lines 427-445).

5. Page 7, line 1-2 please specify what other divalent metals that have been used to test if they are able to change the expression of zitB.

>> We specified the divalent metals we examined. None of Co, Cd, Cu, Fe, Mn, and Ni induced the zitB expression significantly (lines 121-122)

6. Page 7: the authors tested the phenotype of over expressing ZitB. To make their argument completed, it would be helpful to see the phenotype for Δ zitB and Δ zur respectively under the high zinc conditions.

>> We were not able to obtain deletion mutant of zitB after several trials over two years. We therefore examined ZitB function by overexpressing it.

7. Figure 2(c), I wonder if the authors also have measured other metal contents, such as Fe, Mn and Cd? Did they show any changes when comparing WT+vector vs. WT+pZitB?

>> We measured more metals, and found that ZitB overexpression decreased the level of iron also. We were not able to detect Mn and Cd. We included data for iron in the new figure (Fig 2c). Student t-test was done to show p-values. We described this result (lines 137-139, 141-142).

8. I recommend moving Figure S2 to Figure 4 and form another panel there, below the footprinting data. Figure S2 provides very important types of evidence: the peak on the promoter of *zitB* from the ChIP-chip assay, the mapping of the TSS on the *zitB* promoter, and the relative location of the Zur-box motif to -35, -10 region on the *zitB* promoter. These are important features of the major conclusions in this study.

>> We moved Figure S2 to Fig. 4 as recommended.

Reviewer #2 (Remarks to the Author):

The authors report that expression of the zinc exporter, *zitB*, in *Streptomyces coelicolor* is induced by zinc. The authors report that Zur, a transcriptional regulatory protein can repress as well as activate genes involved in zinc-uptake and zinc-export of genes. Activation of genes occurs in two phases with phase one activation occurring at the location of the Zur- box motif, and Phase two involving Zur binding in oligomers upstream of the Zur-box motif.

The manuscript is relatively easy to follow and most experiments are described in detail. Although this paper describes a novel mode of action for Zur, I have some major concerns.

Major comments:

1. Using ChIP-Chip, the authors find ~17,000 potential binding sites for Zur. The authors choose to focus on only the top1% or 172 loci. 17,000 loci is a rather large number. Because this number seems unreasonable, I think it is necessary to include more details on how the analysis was performed to achieve a final list of 17,000 loci and subsequently, what represents the top 1% (i.e. strongest signal?). For example, the authors label the DNA from mock precipitations with Cy3. How were the mock precipitation samples used in the analysis? Were precipitations from WT cells labeled with Cy5 and compared against mock (Cy3) and *zur* deletion samples labeled with Cy5 and compared with mock also? How many individual precipitations were performed (any biological replicates?). Since a large number of putative loci were detected, it appears that maybe the conditions were not optimized. It is also worth mentioning what the breakdown for the locations in the genome that were found putatively

bind Zur (how many of the 17,000 sites are intergenic, etc)?

>> We re-phrased this part of the Results and Methods sections (lines 85-87, 434-439). The original description was somewhat obscure, and could have been misleading.

The genomic regions significantly enriched by Zur binding were identified by TAMALPAIS at $p \leq 0.01$ for a threshold set at the 99th percentile of the log₂ ratio for each chip. Three biological replicates were done. The selected 172 regions (peak sites) were ranked by relative peak intensity, the average of the log₂ ratios of the 10 highest consecutive probes in each region, and were presented as Supplementary Table 1. We removed description about 17,000 loci, since it only served as a pool to select the 172 significant Zur-enriched peaks. We focused on analyzing 172 sites.

2. To identify a motif for Zur, MEME is used. Parameters used for the MEME analysis should be included in the methods. The authors state that the motif that they discover is an improved one. How is it improved? Why do the authors feel it is improved? It seems odd that only 41 out of the 172 loci reside in intergenic regions. Can the authors explain?

>> We used the ZOOPS (Zero or One Occurrence per Sequence) palindrome constraint in MEME analysis. We included this description in the Methods (lines 440-441). The reason we mentioned about improvement is simply because the sequence logo was obtained from larger sets of sequences. We added a new figure (Supplementary Fig. 1) that compares Zur-box consensus sequences, obtained from previous and current studies, as recommended by Reviewer #1.

Currently we cannot explain why there are so many intragenic Zur binding sites. The current reference genome information of *S. coelicolor* contains a number of mis-identified or non-identified ORFs. Therefore, some intragenic binding could turn out to be intergenic. However, as occurs in various other regulator bindings, quite a number of intragenic bindings appear to occur. We hope to understand this phenomenon in the near future.

3. How was Zur purified for EMSA?

>> We added a sentence in the corresponding Methods section about Zur purification with a reference (lines 473-475).

4. It is critical that the authors include a non-specific DNA probe for the EMSA, especially since the ChIP-Chip experiments yielded ~17,000 loci that Zur could potentially bind.

>> In the EMSA experiments, we add an excess amount of poly(dI-dC) as a non-specific competitor, to monitor specific binding of Zur. Under this condition, only the specific binding shifts the probe band. This is confirmed by the data in Supplementary Figure 11c, where the zitB-downstream probe DNA is not bound by Zur at all.

5. The authors pick one gene, zitB, to investigate in more detail. This gene is intriguing because it is an exported and homologs have not been previously reported to be activated by Zur. Since the authors are making the claim that Zur can act as a repressor of genes involved in uptake as well as an activator of genes involved in export, then I think the authors should investigate other potential binding sites where Zur could be acting as an activator. It would be helpful to have global transcript data to compare with the ChIP data to identify genes that are activated vs repressed.

>> We selected the zitB gene to study in further detail because its putative function relates with metal export, unlike other genes of Zur regulon that

encode zinc import and mobilization functions. Finding more genes activated by Zur requires transcriptome and ChIP analyses under both low and high conditions. Incorporation of information about transcription initiation sites will be necessary to locate the position of Zur binding site relative to the promoter -35 element. We hope to pursue along this line in the nearest future.

Minor comments:

1. The format of the references is odd. The text wraps from line to line. This needs to be fixed.

>> We corrected this problem.

2. Figure 2, panel B. What do the pictures showing 2 bands represent? This panel or section should be labeled or explained in Figure legend.

>> The control gel bands refer to ribosomal RNAs. We added the label in the figure.

3. I am not sure that the structural model adds anything. I would remove, especially since in order to generate the model, several parameters and modifications were required.

>> We believe the structural models help to understand how tetrameric Zur can occur on 33-bp DNA probe, different from the way by which dimeric Zur binds to the 25-bp probe. The models show the possibility, which will be tested by structural studies.

Reviewer #3 (Remarks to the Author):

Summary:

The manuscript by Choi et al. describes the role of the *Streptomyces coelicolor* zinc uptake regulator (ZUR) in the transcriptional regulation of *znuA*, a zinc import system, and *zitB*, a zinc export system. The authors discern two phases of ZUR-dependent transcriptional regulation. At sub-femtomolar zinc concentrations (phase I), *znuA* is up regulated and *zitB* is repressed. At over micromolar zinc concentrations (phase II), *znuA* remains (fully) repressed and *zitB* is transcriptionally activated. Subsequent analyses focus on the oligomeric binding of ZUR dimers to the upstream region of *zitB* during phase II, which leads to its transcriptional activation. The key findings in this manuscript as described by the authors are 1) the transcriptional activation of a zinc efflux system by ZUR, 2) the broad concentration range at which ZUR modulates expression of its targets and 3) the oligomerisation of ZUR dimers upwards from the *zitB* ZUR box.

Major Comments:

- The abstract, introduction and results of this manuscript are written such that the reader is under the impression that transcriptional activation by Zur is novel, particularly its role as an activator of metal efflux systems. However, the authors fail to appropriately introduce ZUR and in general the FUR family of regulators, thereby, in part, misleading the reader about the novelty of their work. There are many

studies that describe activation by these types of regulators (e.g. doi: 10.3389/fcimb.2013.00059, doi: 10.1128/JB.00166-16, doi: 10.1093/nar/gkn328, doi: 10.1128/JB.01091-12). Their statement in the introduction “In all systems reported so far, the depletion.....for the specific metal.” is incorrect considering findings from Zur in *Xanthomonas campestris*. In the discussion the authors briefly discuss Zur-dependent activation of a putative metal-efflux system in *X. campestris*, but then report that this form of regulation is distinct to that seen in *S. coelicolor*, as the relevant Zur binding site was distinct from those in the *X. campestris* Zur box regulating the zinc uptake systems. However, Zur-dependent activation was also observed in the meningococcus and in this study (Pawlik et al. 2012 J Bact.) the Zur binding site was homologous to that seen for the Zur binding sites identified upstream zinc uptake systems. Overall, it is not unexpected that Zur-dependent activation of a zinc efflux system has been identified and the lack of appropriate introduction of available literature exaggerates the claim of novelty in regards to this observation in the manuscript by Choi et al..

>> One major novelty of this work is in demonstrating that Zur regulates both the uptake/mobilization genes and export genes in response to a wide range of zinc concentration changes, by binding to Zur-box sequences. The meningococcal Zur was proposed to activate two genes encoding putative alcohol dehydrogenase and NosR-related protein, respectively, based on transcriptome analysis (Pawlik et al., 2012). The *Xanthomonas* Zur was reported to activate a gene encoding a putative metal exporter based on transcriptome analysis and in vitro binding studies (Huang et al., 2018). In the *Xanthomonas* system, however, Zur did not recognize the Zur-box in the putative export gene. Therefore, even though there are reports that Zur activates some genes, demonstration of zinc-dependent activation of the experimentally characterized exporter gene via binding to the Zur-box consensus has been made only in this study.

As the authors pointed out, we included the comparison of our findings with related works, and elaborated discussion on the dual role of Zur and Fur as a repressor and an activator, referring to the relevant papers (lines 313-314, 337-338, 348-355).

In the introduction, we modified the sentence, “In all systems reported so far, the depletion.....for the specific metal.”, to include findings from *Xanthomonas* Zur study (lines 27-30).

- Although the oligomerization of Zur upstream *zitB* is an interesting finding, the oligomerization of Fur along its target DNA has previously been shown (e.g. Escolar et al. J Biol Chem. 2000; Fréchon and Le Cam Biochem Biophys Res Commun. 1994; Le Cam et al. Proc Natl Acad Sci U S A. 1994). Even studies presented by the authors themselves (Shin et al. 2007) have indicated oligomerisation of *S. coelicolor* Zur on sites of transcriptional repression (EMSA and DNaseI footprint). Although analytical ultracentrifugation did not reveal in-solution, DNA-target free, oligomerisation of Zur, this was not performed in the presence of high zinc concentrations, which should be considered. Overall, the finding described by Choi

et al. are not entirely novel for this family of metalloregulators.

>> We did not observe any oligomer formation of Zur in the absence of DNA at high protein and high zinc conditions (Supplementary Figure 8), unlike Fur which can oligomerize in the absence of DNA. We added this information in the revision (lines 216-219), and referred to the Fur oligomerization studies by Le Cam et al. (lines 346-348).

- Greater emphasis on the mechanisms by which zitB is upregulated would significantly enhance this manuscript. The following questions follow up from previous findings on oligomerisation by Fur-like proteins and addressing these should be considered by the authors.

1) Does high zinc affect ionic strength and subsequently binding of Zur to its targets?

>> If high zinc effect is due to changes in ionic strength, we may see similar activation with other metals as well. However, among divalent metals we examined (Zn, Cd, Co, Fe, Ni, Mn, Cu), only zinc induced the zitB gene. Therefore, the activation is very zinc-specific.

2) Does Zur undergo structural changes in a high zinc environment?

>> We observed electrophoretic mobility of Zur at high zinc up to 0.5 mM, and found no hint of oligomeric formation in the absence of DNA.

3) What are the characteristics of the DNA sequence upwards from the zitB zur box? (Are there repeat sequences such as described in Escolar et al. 2000 jbc? Are there any significant secondary structures?)

>> We were not able to pinpoint any specific feature in the upstream sequences. It will be interesting to find any weak binding sites that may facilitate oligomer binding, which we would like to pursue further.

- The data in Fig. 3 should be backup by an alternative method such as qRT-PCR analysis, whereby the same RNA samples are used to analyse znuA and zitB expression. The data presented in fig 3 is from different S1 mapping analyses and the number of replicates vary greatly (n=3, 6 or 11). Data for znuA and zitB expression in Fig 3B should be corrected to untreated media (without TPEN or Zn supplementation). Standard deviations should be included in Fig 3B.

>> As suggested, we performed qRT-PCR analysis to confirm the S1 results. The qRT-PCR produced similar fold change values, and the data were presented as Supplementary Figure 4.

As recommended by the reviewer, we presented Fig. 3a data as bar graphs with standard deviations, instead of the gel photos with numbers. The fold-changes of znuA and zitB in Figure 3b were corrected relative to the untreated values.

- How is it possible that *zitB* is nearly 2-fold down-regulated (0.6-fold) under TPEN stress in the *delta-zur* strain? This level of down-regulation is similar to that seen in the wt strain (0.5-fold). The claim that down-regulation of *zitB* under TPEN stress is Zur-dependent does not seem valid based on these results.

>> It appears that there exists some additional metal-dependent regulation other than Zur that acts on the *zitB* promoter under low zinc condition. The decreased (by ~80%) basal level of *zitB* RNA due to *zur* mutation in non-treated sample demonstrates clearly that Zur activates *zitB* under this low zinc condition. We mentioned about this possibility in the revision (lines 122-127).

- Why were nickel and copper not tested in the Zur-dependent activation of *zitB* (Fig. 1D)?

>> We tested nickel and copper as well as iron and manganese and found that they do not induce *zitB* expression. We added this information in the revision (lines 121-122)

- The *zitB* band in Cd stressed *delta-zur* cells appears a lot more intense than that seen for untreated *delta-zur*, but is not supported by the average? Furthermore, Cd appears to cause significantly down-regulation of *zitB* in the wt, but again this is not supported by the averaged numbers. The metal content of Cd, Co and Ni stressed cells should be determined by ICPMS to gain insight into their potential effect on Zn accumulation and subsequently *zur*-mediated activation of *zitB*, in isolation these stressed conditions are ineffectual.

>> As stated in response to reviewer 1, the original gel photo we presented was not the representative one that matches with the quantified values. From repeating experiments with wider repertoire of divalent metals, we replaced it with newly obtained gel photo with quantified values. The new Fig. 1d clearly shows that zinc is the only inducing metal, and that in *delta-zur* mutant the *zitB* gene is not induced by any metal.

- Analysis of the physiological function of ZitB, should also be confirmed by gene deletion. The role of ZitB in Co, Ni and Cd export under the relevant metal stress should be examined by ICP-MS using this mutant and compared to the wt and overexpressing strain. Only testing metal accumulation under low metal abundance may not reveal its true role in metal efflux.

>> We were not able to obtain deletion mutant of *zitB* after several trials over two years. We therefore examined ZitB function by overexpressing it.

- This manuscript would greatly benefit from examinations into the broader application of their findings. *S. coelicolor* is a model organism, its findings have to be discussed in broader context.

1) Were other candidates of Zur-dependent transcriptional activation identified within the genome?

>> We were not able to identify other genes that could be activated by Zur binding. As stated in response to reviewer 1, finding more genes activated by Zur requires transcriptome and ChIP analyses under both low and high zinc conditions. Incorporation of information about transcription initiation sites will be necessary to locate the position of Zur binding site relative to the promoter -35 element. We hope to pursue along this line in the nearest future.

2) Are regions with similar GC% or sequence identity found upwards from other transcriptionally activating Zur-boxes?

>> We do not have other candidate genes to compare at this moment.

3) Can similar targets be found in other bacterial genera/species?

>> We found that the Zur-box sequence is conserved upstream of the zitB homologs in other actinomycetes. Therefore, it is likely that similar zinc homeostatic regulation may occur in other actinomycetes such as Mycobacterium and Corynebacterium species. We stated this in the Discussion section (lines 326-329).

• Although Supplementary figure 8 shows that the addition of zinc increases Zur binding at the Zur box, it does not show binding further upwards from the Zur box under high zinc conditions as compared to untreated conditions. Please discuss this.

>> With the PCR-generated probes ranging in length from 65 to 93 bp overlapping by 20 bp with neighboring probes, the resolution may not be high enough to see the change in the boundary of the peak. ChIP experiments combined with exonuclease treatment (ChIP-exo) may be able to produce higher resolution of in vivo Zur binding pattern.

• Statistical analyses are missing from most presented data. Often already in the form of standard deviation or standard error, but also when comparing different samples, e.g. t-test, ANOVA, etc.. This should be addressed for Fig. 1A, 1D, 2B, 2C, 3A, 3B, 5C, and Supplementary Fig. 3, 4B, 8.

>> We applied t-test and added p-value information in Figs. 1D, 2B, 2C, 3A, 5C and Supplementary Figs. 4 and 12.

• I recommend presenting the data as bar graphs instead of gels/blots with numbers underneath. This allows for easier interpretation of the data by the reader and statistical comparisons between samples to support some of the claims made. This concerns Fig. 1A, 1D, 2B, 3A, 5C.

>> We changed Fig. 3a, which contain largest number of gel photos, to bar graphs with error bars. For other figures (1a, 1d, 2b, and 5c), bar graphs take too much space. We preferred to show gel/blot photos with the numbers.

- Please deposit the ChIP-chip data in GEO or other suitable databases.

We deposited the data on GEO, and specified the accession number GSE95760 (<https://www.ncbi.nlm.nih.gov/geo/query/acc.cgi?acc=GSE95760>) in the methods (lines 442-445).

- The experiments have been described in sufficient detail in the methods section.

>> We added some further details in the methods section, such as Zur purification and data analysis of ChIP-chip assay. Methods for newly added experiments such as in vitro transcription assay were added.

- The manuscript contains various grammatical errors.

Specific comments:

Title: Consider rephrasing the title, e.g. “Zinc-dependent regulation of zinc import and export genes by Zur.”

>> We changed the title as recommended.

Abstract Page 2 Line 6: Insert “by Zur” after “activation”

>> changed as suggested.

Introduction Page 2 first paragraph line 20: “very tightly” and “within a narrow range” means the same.

>> changed as suggested.

Page 3 Line 4-6: As mentioned above, the statement “In all systems reported.....specific metal.” is an overstatement.

>> We revised this sentence as stated above.

Page 3: Include more detail about the role of fur and zur as transcriptional activators and describe oligomerisation of Fur.

>> We added this information in the Discussion section.

Page 3 line 18: Include “the” after “ZntR of”

>> Yes

Page 3 line 19: Include “the” before “ArsR”

>>Yes

Page 4 line 5-6: This sentence “zinc-binding antibiotic.....zinc starvation.” seems very random, consider removing.

>> Yes, we removed the sentence.

Page 5 line 10: Include “the” before “cell”.

Yes

Page 5 line 14-15: Define Y-axis in Fig 1B.

>> We labeled Y-axis (ChIP peak intensity) and defined it in the legend to Fig. 1B.

Page 5 line 14-15: Instead of only stating 1%, is there a possibility of grading the spread of high represented sites? Is there a p-value cut-off that can be used instead?

>> We rephrased this part of the text, as recommended also by reviewer 2. (lines 85-87)

The p-value cut-off of 0.01 and the threshold setting of 99th percentile were mentioned in the Methods section (lines 434-439). 172 Zur-enriched sites were ranked by relative peak intensity, and were presented in Supplementary Table 1.

Page 5 line 21: This sentence requires rewriting “improved version” and “previous one” is too vague.

>> We provided more information about Zur-box consensus determinations from previous studies in Supplementary Figure 1, as also suggested by Reviewer 1.

Page 5 line 21: Correct “share” to “shares”

>> Yes

Page 5 line 21-22: define “some features”

>> We elaborated information about the computational Zur-box sequence obtained from the zunA genes of 17 actinobacterial genomes (Schroder et al., 2010), to make better comparison, in the text and in Supplementary Fig. 1 (lines 90-96).

Page 6 line 3: Do these 41 loci also represent the best hits? Present p-values in a table.

>> Yes. Zur-enriched regions with P-values of less than 0.01 were only considered.

Page 6 line 7: See comment “Page 5 line 14-15” regarding 1% cut-off.

>> We took care of this issue as stated above.

Page 6 line 16: Naming a gene (zitA) of which the function has not been determined in this manuscript does not appear appropriate to me.

>> We rephrased this sentence to state that SCO1310 clusters closely with the zitA gene of Mtb (lines 111-113).

Page 6 line 23: See major comments regarding near 2-fold down regulation of zitB under TPEN stress in the delta-zur strain.

>> We added explanation as stated above.

Page 7 line 3-17: See major comments regarding zitB gene deletion and testing of metal stress conditions.

>> We responded above.

Page 7 line 10: delete “the” before zitB-expression.

>> **Yes.**

Page 8 line 2-3: “When treated with TPEN, the.....znuA expression.” Considering the down-regulation of zitB in the delta-zur strain under TPEN stress, the data does not convincingly support the role of zur in down-regulation of zitB under severe zinc starvation. This has already been pointed out in the major comments.

>> **We responded to this comment above. There could be additional metal-dependent regulation under severe zinc starvation.**

Page 8 line 1-5: Fig. 3a is too messy, as mentioned in the major comments, these findings have to be backed up by an alternative method in which zitB and znuA transcription can be determined from the same samples. All samples should be corrected against untreated cells.

>> **We confirmed the S1 values with qRT-PCR assay as stated above (Supplementary Fig. 4). Fig. 3a was presented as bar graphs. The fold change was re-scaled relative to the untreated samples.**

Page 8 line 20: Please generate a single figure in which the data of Supp Fig 3 and Fig 3a can all be displayed.

>> **If we combine those figures, the graph will become too complicated to express the main finding. We hope to present the data in separate figures.**

Page 9 lines 5-7: Please put Kd values into context in terms of bioinformatics of the Zur box (p-values) and ChIP-chip data.

>> **The Kd values from EMSA demonstrate similar binding affinities of Zur to both zitB and znuA Zur-box sequence. We related this similar Kd values with similarity in peak intensity in ChIP-chip data, by adding a sentence (lines 202-204); “The similarity in Kd values is consistent with the observation that Zur-binding was enriched with similar peak intensities at the znuA and zitB sites in ChIP-chip analysis (Supplementary Table 1).”**

Page 11 line 10: Supplementary Fig. 5, this should refer to supp. Fig 6.

>> **We corrected for Figure numberings.**

Page 11 line 13: Include “a” after “at”

>> **Yes**

Page 11 line 16: Include “a” after “when”

>> **Yes**

Page 11 line 19-20: Please expand on the multi-angle light scattering, I have not seen this described anywhere in the manuscript.

>> **We removed this sentence.**

Page 12 line 18: Include “the” before “zitB”

>> **Yes**

Page 14 line 13-15: Please discuss the level of conservation to the -138 region in other actinobacteria.

>> The upstream sequence did not show any pronounced conservation. Further systematic studies to identify as yet uncharacterized sequence features are in need. We mentioned about this in the text (lines 329-332).

Page 14 line 20: Please correct to: "..., Fur and Zur are known to play a dual role."

>> Yes

Page 15 line 7-10: Please include report by Pawlik et al. 2012 J Bact.

>> Thank you for drawing our attention to this work. We did.

Page 17 line 6: Please define "Standard protocols"

>> Yes, we did.

Page 17 line 8: rephrase "strong promoter ermEp fragment".

>> We changed it to "strong ermEp promoter"

Page 18 line 9: Include "The" before "Chromatin"

>> Yes

Page 18 line 15: Include "the" before "top 1%"

>> We rephrased this part of Methods section.

Page 18 line 16: Remove "by" after "0.01"

>> Same as above.

REVIEWERS' COMMENTS:

Reviewer #1 (Remarks to the Author):

The authors have taken all suggestions, provided additional data and addressed each of the concerns raised in the review comments.

1. They did transcription run-off assays and measured *znuA* and *zitB* expression under various zinc concentrations;
2. Made a table list the Top 1% peaks and corresponding genes from the Chip-seq;
3. Repeated the gels to make the intensities of the bands well match with their quantification results;
4. Fixed all the other minor concerns.

This manuscript an important addition to our understanding of cellular zinc regulation mechanisms and I recommend publication

Reviewer #2 (Remarks to the Author):

I have looked at the point by point response letter and the revised manuscript. I feel that the points raised have been satisfactorily addressed and this paper is nicely put together.

Reviewer #3 (Remarks to the Author):

The authors have made a significant effort in following up on the reviewer's comments and have improved their revised manuscript. I have a few comments remaining.

Line 26: Insert comma after "(Phase I)"

Line 36: Delete blank line

Line 48: Insert "However" before "In *Xanthomonas campestris*"

Lines 78-79: Start new paragraph.

Lines 121-122: Please compare ZUR box E-value with, quantitative DNA enrichment from ChIP-Chip analyses.

Lines 165-166: Mention iron.

Figure 3A: *zitB* should be in italics.

Line 228: Please explain in the context of E-values and include Supplementary Figure 7.

Line 290: Explain in the context of the ChIP-Chip experiment. This preferential binding to the upstream region was not observed in Suppl. Fig. 3., which showed even distribution from the ZUR box.

Response to Reviewers' comments.

REVIEWERS' COMMENTS:

Reviewer #1 (Remarks to the Author):

The authors have taken all suggestions, provided additional data and addressed each of the concerns raised in the review comments.

1. They did transcription run-off assays and measured znuA and zitB expression under various zinc concentrations;
2. Made a table list the Top 1% peaks and corresponding genes from the Chip-seq;
3. Repeated the gels to make the intensities of the bands well match with their quantification results;
4. Fixed all the other minor concerns.

This manuscript an important addition to our understanding of cellular zinc regulation mechanisms and I recommend publication

>> Thank you very much.

Reviewer #2 (Remarks to the Author):

I have looked at the point by point response letter and the revised manuscript. I feel that the points raised have been satisfactorily addressed and this paper is nicely put together.

>> Thank you very much.

Reviewer #3 (Remarks to the Author):

The authors have made a significant effort in following up on the reviewer's comments and have improved their revised manuscript. I have a few comments remaining.

>> The line numbers quoted by the referee are different from what we submitted. In the following response, we located the position, and made changes. New line numbers in the finally revised manuscript were indicated, when necessary.

Line 26: Insert comma after "(Phase I)"

>> Yes, we did.

Line 36: Delete blank line

>> **Yes, we did.**

Line 48: Insert “However” before “In *Xanthomonas campestris*”

>> **We changed as recommended.**

Lines 78-79: Start new paragraph.

>> **It was taken care of by the editor.**

Lines 121-122: Please compare ZUR box E-value with, quantitative DNA enrichment from ChIP-Chip analyses.

>> **We added E-value information of Zurbox sequence extraction by MEME analysis (3.9e-233) in the legend to Fig. 1c as well as in the Supplementary Figure 1.**

Lines 165-166: Mention iron.

>> **We did.**

Figure 3A: *zitB* should be in italics.

>> **We italicized the gene name.**

Line 228: Please explain in the context of E-values and include Supplementary Figure 7.

>> **We included description about Supplementary Figure 7 (lines 214-215). We preferred not to explain further about E-values here.**

Line 290: Explain in the context of the ChIP-Chip experiment. This preferential binding to the upstream region was not observed in Suppl. Fig. 3., which showed even distribution from the ZUR box.

>> **To clarify description, we changed the word ‘ChIP results’ in line 266 to ‘ChIP-qPCR results’. To our opinion, direct comparison of ChIP-qPCR results with ChIP-chip data in Suppl. Fig. 3 is not meaningful, since the resolution of ChIP-chip data is relatively poor, and not comparable to the ChIP-qPCR data.**

Thank you for careful suggestions to improve our manuscript.